# YAP/TAZ deficiency reprograms macrophage phenotype and improves infarct healing and cardiac function after myocardial infarction

**Masum M. Mia[1], Dasan Mary Cibi[1], Siti Aishah Binte Abdul Ghani[1], Weihua Song[2], Nicole Tee[2], Sujoy Ghosh[1], Junhao Mao[3], Eric N. Olson[4], Manvendra K. Singh[1,2]***

1 Cardiovascular and Metabolic Disorders Program, Duke-NUS Medical School Singapore. Singapore,
2 National Heart Research Institute Singapore, National Heart Centre Singapore, Singapore, 3 Department of Molecular, Cell and Cancer Biology, University of Massachusetts Medical School, Worcester, Massachusetts, United States of America, 4 Department of Molecular Biology, University of Texas Southwestern Medical Center, Dallas, Texas, United States of America

* manvendra.singh@duke-nus.edu.sg

**Data Availability Statement:** All relevant data are within the paper and its Supporting Information files. Raw RNAseq data set has been submitted to

## Abstract

Adverse cardiac remodeling after myocardial infarction (MI) causes structural and functional changes in the heart leading to heart failure. The initial post-MI pro-inflammatory response followed by reparative or anti-inflammatory response is essential for minimizing the myocardial damage, healing, and scar formation. Bone marrow–derived macrophages (BMDMs) are recruited to the injured myocardium and are essential for cardiac repair as they can adopt both pro-inflammatory or reparative phenotypes to modulate inflammatory and reparative responses, respectively. Yes-associated protein (YAP) and transcriptional coactivator with PDZ-binding motif (TAZ) are the key mediators of the Hippo signaling pathway and are essential for cardiac regeneration and repair. However, their functions in macrophage polarization and post-MI inflammation, remodeling, and healing are not well established. Here, we demonstrate that expression of YAP and TAZ is increased in macrophages undergoing pro-inflammatory or reparative phenotype changes. Genetic deletion of *YAP/TAZ* leads to impaired pro-inflammatory and enhanced reparative response. Consistently, YAP activation enhanced pro-inflammatory and impaired reparative response. We show that YAP/TAZ promote pro-inflammatory response by increasing interleukin 6 (*IL6*) expression and impede reparative response by decreasing Arginase-I (*Arg1*) expression through interaction with the histone deacetylase 3 (HDAC3)-nuclear receptor corepressor 1 (NCoR1) repressor complex. These changes in macrophages polarization due to *YAP/TAZ* deletion results in reduced fibrosis, hypertrophy, and increased angiogenesis, leading to improved cardiac function after MI. Also, YAP activation augmented MI-induced cardiac fibrosis and remodeling. In summary, we identify YAP/TAZ as important regulators of macrophage-mediated pro-inflammatory or reparative responses post-MI.

GEO and accession number (GSE158889) is
provided in the Materials and methods section.

**Funding:** This work was supported by funds from
Duke-NUS Medical School Singapore and the Goh
foundation and a Singapore National Research
Foundation (NRF) fellowship (NRF-NRFF2016-01)
to M.K.S. The work in Mao lab was supported by
NIH grant R01DK099510. The funders had no role
in study design, data collection and analysis,
decision to publish, or preparation of the
manuscript.

**Competing interests:** The authors have declared
that no competing interests exist.

**Abbreviations:** 2D, 2-dimensional; Acta2, smooth
muscle actin alpha 2; APC, Allophycocyanin; Arg1,
Arginase-I; BMDMs, bone marrow–derived
macrophages; Ccl2, C–C motif chemokine ligand 2;
Ccr7, C-C chemokine receptor type 7; Cd206,
cluster of differentiation 206; ChIP, chromatin
immunoprecipitation; CD11b-FITC, cluster of
differentiation 11B-fluorescein isothiocyanate;
Col1α1, collagen type I α 1; DMEM, Dulbecco's
Modified Eagle Medium; echo, echocardiographic;
Egr2, early growth response 2; Emcn, endomucin;
Erk1/2, extracellular signal-regulated kinase ½;
FAC, fractional area change; FACS, fluorescence-
activated cell sorting; FDR, false discovery rate;
FGF2, fibroblast growth factor 2; Fizz1, found in
inflammatory zone 1; FS, fractional shortening;
Gapdh, glyceraldehyde-3-phosphate
dehydrogenase; HDAC3, histone deacetylase 3;
IDT, Integrated DNA Technologies; IFNγ, interferon
gamma; IgG, immunoglobulin G; IKKαβ, inhibitor
of nuclear factor kappa-αβ-kinase; IL, interleukin;
IRES, internal ribosome entry site; IRF5, interferon
regulatory factor 5; JNK, c-Jun N-terminal kinase;
KEGG, Kyoto Encyclopedia of Genes and Genomes;
LAD, left anterior descending; Lats, large tumor
suppressor kinase; Lox, lysyl oxidase; LPS,
lipopolysaccharide; LVEDV, left ventricular end-
diastolic volume; LVEF, left ventricular ejection
fraction; LVESV, left ventricular end-systolic
volume; LVIDed, left ventricular internal diameter at
end-diastole; LVIDes, left ventricular internal
diameter at end-systole; LV Mass, left ventricular
mass; MAP, mitogen-activated protein; MCP-1,
monocyte chemoattractant protein-1; MI,
myocardial infarction; Mmp9, matrix
metallopeptidase 9; MPO, myeloperoxidase; MPO
+, myeloperoxidase positive; MyD88, myeloid
differentiation primary response 88; Myh7, myosin
heavy chain 7; NCoR, nuclear receptor
corepressor; NCoR1, nuclear receptor corepressor
1; NF-kB, nuclear factor kappa B; NO, nitric oxide;
Nos2, nitric oxide synthase 2; Nppa, natriuretic
peptide A; Nppb, natriuretic peptide B; NuRD,

# Introduction

Myocardial infarction (MI) and the risk of subsequent heart failure are a major global health
burden with significant mortality and morbidity. MI occurs from obstructed coronary artery,
causing inadequate oxygen and nutrient supply to the heart muscle or myocardium, leading to
tissue hypoxia and cell death. Cardiomyocyte death triggers an acute inflammatory response
through macrophage recruitment to the infarcted area, which ultimately activates reparative
pathways necessary for preventing further loss of cardiomyocytes, fibrotic scar formation, and
restoring tissue integrity [1–3]. Majority of these macrophages are derived from differentiated
peripheral blood monocytes from the bone marrow and spleen. These macrophages orches-
trate the myocardial repair response by secreting pro-/anti-inflammatory, pro-angiogenic, and
pro-reparative factors, as well as removing dead cells through phagocytosis [4–9]. The diverse
functions of macrophages in infarcted hearts are partially attributed to their ability to adopt
different phenotypes and polarization status in response to environmental stimuli. Based on
their inflammatory properties, macrophages are classified into 2 major groups: classically acti-
vated pro-inflammatory macrophages and alternatively activated anti-inflammatory/repara-
tive macrophages [10–12]. Reparative macrophages can be further subdivided into M2a, M2b,
M2c, and M2d [13]. Pro-inflammatory macrophages, responsible for stimulating an inflamma-
tory response, dominate the heart at day 1 to 3 post-MI and secrete high level of pro-inflamma-
tory cytokines and chemokines such as interleukin (IL)-6, IL-1β, IL-12β, Rantes, monocyte
chemoattractant protein-1 (MCP-1), and tumor necrosis factor alpha (TNFα). In contrast,
reparative macrophages, responsible for preserving the structural integrity of the injured ven-
tricle by promoting cardiac repair through myofibroblasts induction, collagen deposition, and
neovascularization, dominate the heart at day 5 to 7 post-MI and secrete high level of anti-
inflammatory cytokines and angiogenic factors such as IL-10, vascular endothelial growth fac-
tor (VEGF), fibroblast growth factor 2 (FGF2), and transforming growth factor beta (TGFβ)
[3,11,14–18]. Among other markers, Arginase-I (Arg1), the enzyme that is involved in l-argi-
nine/nitric oxide (NO) metabolism, mannose receptor CD206, chitinase-like lectin YM1, and
resistin-like secreted protein FIZZ1 are also associated with the reparative phenotype [11,19–
23]. A fine balance between pro-inflammatory and reparative macrophage response is essential
for optimal repair as enhanced or persistent pro-inflammatory response can delay the repara-
tive macrophage-mediated repair response and exacerbate adverse ventricular remodeling.
Growing evidence suggests that a shift from pro-inflammatory to reparative phenotype has a
positive effect on cardiac repair and function post-MI [24–28]. Thus, identifying new factors
involved in modulating pro-inflammatory/reparative responses will provide novel therapeutic
targets that may prevent adverse cardiac remodeling and heart failure post-MI.

Yes-associated protein (YAP) and transcriptional coactivator with PDZ-binding motif
(TAZ) are the main downstream transcriptional regulators of the Hippo signaling pathway.
Extensive preclinical studies have established the role of the Hippo pathway components in
cardiac development, regeneration, and cellular homeostasis. The growing evidence suggests
that they are also involved in regulating biological processes other than growth and develop-
ment, such as immune response [29–40]. For example, Hippo signaling components modulate
tumor microenvironment through interactions with immune cells such as macrophages. Simi-
larly, YAP-driven CXCL5 produced by prostate cancer cells can lead to infiltration of the mye-
loid-derived suppressor cells to the tumor site [30]. YAP functions downstream of PRKCI
oncogene and induces TNFα expression, which promotes myeloid-derived suppressor cells
and inhibits cytotoxic T cell infiltration in ovarian carcinomas [31]. YAP also directs the
recruitment of tumor-associated macrophages, essential for immune evasion and tumorigene-
sis [32,33]. Similar to YAP, TAZ also promotes inflammatory cytokine production and

nucleosome remodeling and deacetylase; P4ha1, prolyl 4-hydroxylase subunit alpha-1; Plod2, procollagen-lysine,2-oxoglutarate 5-dioxygenase 2; PMs, peritoneal macrophages; qRT-PCR, real-time quantitative reverse transcription PCR; SD, standard deviation; SEM, standard error mean; siRNA, short interfering RNA; SMRT, silencing mediator of retinoic and thyroid receptor; SV, stroke volume; Tagln, transgelin; Tak1, transforming growth factor beta-activated kinase 1; TAZ, transcriptional coactivator with PDZ-binding motif; TBSs, Tead-binding sequences; TBST, TBS containing 0.1% Tween; Tead, Transcriptional enhanced associate domain; TGFβ, transforming growth factor beta; Timp1, tissue inhibitor matrix metalloproteinase 1; TLR4, Toll-like receptor 4; TNFα, tumor necrosis factor alpha; Trp53, tumor protein 53; TTE, transthoracic echocardiography; UNT, untreated; Vcam1, vascular cell adhesion molecule 1; VEGF, vascular endothelial growth factor; VP, verteporfin; WGA, wheat germ agglutinin; YAP, yes-associated protein; YAP/TAZ-dKO, YAP/TAZ double knockout.

macrophage infiltration [34]. The Hippo pathway components also modulate the immune response against viral or bacterial infections [29,35–39]. Apart from directing the innate immune response, the Hippo pathway components also modulate the adaptive immune responses in multiple pathological conditions. For example, we recently demonstrated that YAP/TAZ expression in the epicardium is essential not only for coronary vasculature development but also for limiting the inflammatory and fibrotic response during post-MI recovery phase [41,42]. Despite the well-established role of Hippo signaling components in non-immune cells regulating the inflammatory response, very little is known about its functions in immune cells [43–45]. The role of Hippo signaling components in immune cells, particularly in macrophage polarization and macrophage-mediated cardiac inflammation, remodeling, and repair response post-MI is not well established.

In the present study, we identified YAP/TAZ as essential regulators of macrophage polarization and functions. In response to both pro-inflammatory or reparative stimuli, YAP and TAZ expressions are increased in macrophages. Knockdown or conditional genetic deletion of *YAP/TAZ* in macrophages leads to decreased expression of pro-inflammatory genes and increased expression of anti-inflammatory/reparative genes. Our results demonstrate that YAP/TAZ act as an activator in pro-inflammatory macrophages while behaving as a repressor in reparative macrophages. YAP/TAZ promote pro-inflammatory phenotype by directly regulating interleukin 6 (*IL6*) promoter activity or through the p38-dependent MAPK pathway. While during reparative phenotype, YAP/TAZ repress *Arg1* expression by binding to its promoter and recruiting histone deacetylase 3 (HDAC3)-nuclear receptor corepressor 1 (NCoR1) repressor complex. We further demonstrate that *YAP/TAZ*-deficient mice show improved cardiac remodeling and function after MI, evident by reduced cardiac fibrosis, reduced cardiac hypertrophy, and improved tissue angiogenesis. Consistently, overexpression of a constitutively active YAP mutant (YAP^5SA) showed opposite effects on macrophage polarization leading to increased fibrosis and adverse remodeling of the heart post-MI. Together, our findings demonstrate that YAP/TAZ modulate post-MI inflammatory and reparative response by regulating macrophage polarization.

## Results

### YAP/TAZ expression is enhanced in pro-inflammatory macrophages

To determine whether YAP/TAZ are involved in macrophage polarization, we treated mouse bone marrow–derived macrophages (BMDMs) and peritoneal macrophages (PMs) with lipopolysaccharide (LPS)/interferon gamma (IFNγ) (pro-inflammatory macrophage-stimuli) for 0 to 12 hours and prepared protein extracts for whole cell lysate to perform western blot analysis. In BMDMs, the expression of YAP was significantly enhanced after 6 hours of stimulation and remained steady for 12 hours. Similarly, LPS/IFNγ treatment also increased YAP phosphorylation at S127 in BMDM cells (Fig 1A). In PMs, YAP expression was significantly increased after 6 hours post-treatment and remained steady for 12 hours (Fig 1B). Similar to YAP, TAZ expression was also increased after 6 hours of stimulation with LPS/IFNγ and remained steady for 12 hours (Fig 1A and 1B). The pro-inflammatory macrophage phenotype was confirmed by the presence of pro-inflammatory marker IL6 (Fig 1A and 1B). Since YAP/TAZ act in the nucleus, we furthermore measured the nuclear presence of YAP/TAZ together with cytoplasmic expression of pYAP in BMDMs after LPS/IFNγ treatment for 8 and 12 hours. Western blot analysis revealed a significant increase of YAP, TAZ, and pYAP protein levels with pro-inflammatory macrophage-stimuli (S1 Fig). To determine whether LPS/IFNγ treatment resulted in the formation of pro-inflammatory and not reparative phenotype, we also measured the expression of reparative markers and observed no significant change or decrease in

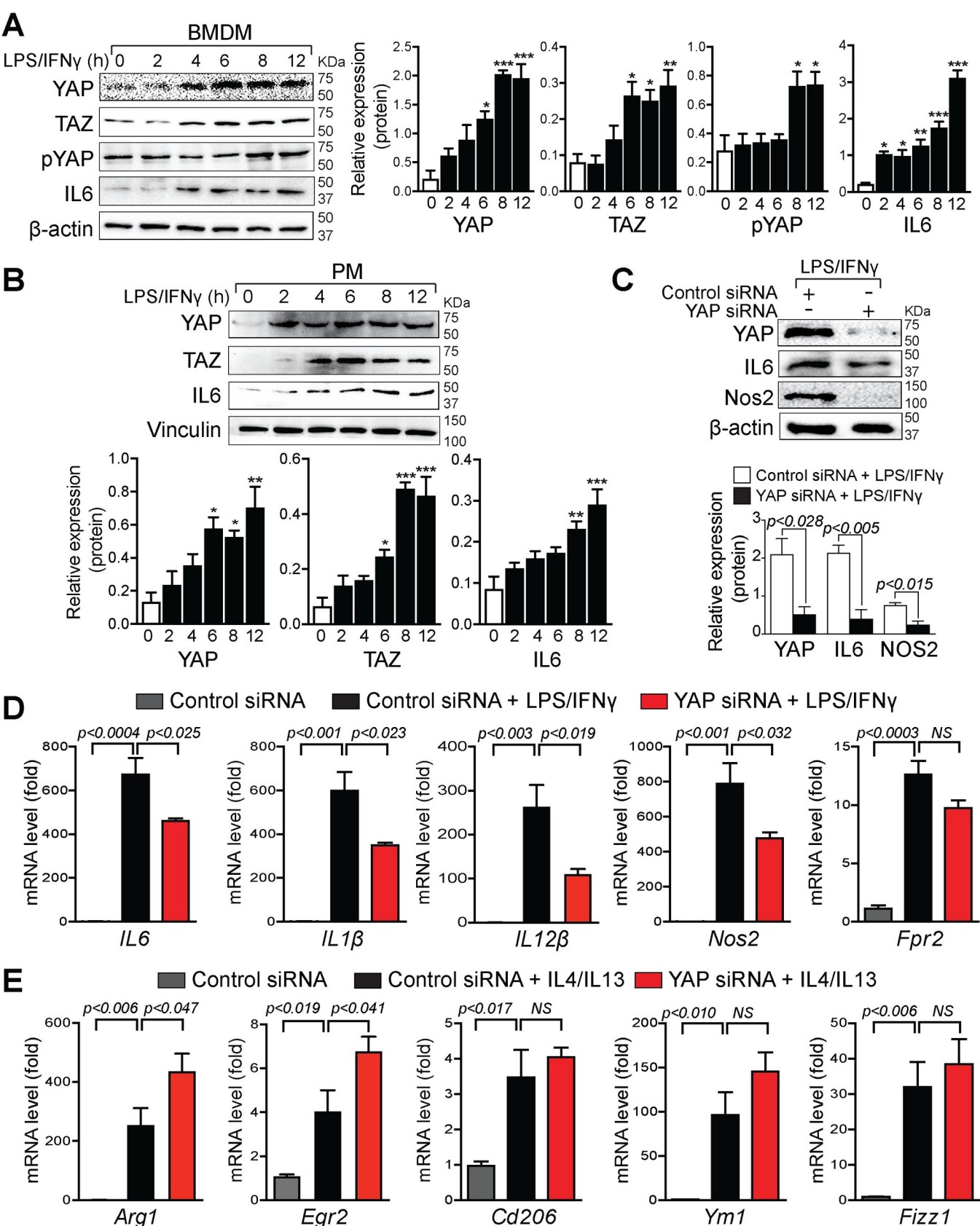

**Fig 1. YAP/TAZ expression is enhanced in pro-inflammatory macrophages.** (A and B) Western blot analysis for YAP, pYAP (S127), TAZ, and IL6 was performed using total lysates from wild-type BMDMs (A) and PMs (B) treated with LPS/IFNγ for 0 to 12 hours as indicated. β-actin or vinculin is shown as a loading control. The relative protein expression was quantified. (C–E) BMDMs were isolated from wild-type mice and transfected with control or *YAP* siRNA for 72 hours, followed by LPS/IFNγ or IL4/IL13 stimulation for 16 hours. Cell lysates were prepared for western blot and qRT-PCR analysis. (C) Western blot analysis for YAP, IL6, and Nos2 was performed using total lysates from wild-type BMDMs transfected with control or YAP siRNA. β-actin is shown as a loading control. The relative protein expression was quantified. (D) qRT-PCR for pro-inflammatory marker genes *IL6*, *IL1β*, *IL12β*, *Nos2*, and *Fpr2* using RNA isolated from wild-type BMDMs transfected with control or YAP siRNA and stimulated with LPS/IFNγ. (E) qRT-PCR for reparative/anti-inflammatory marker genes *Arg1*, *Egr2*, *Cd206*, *Ym1*, and *Fizz1* using RNA isolated from wild-type BMDMs transfected with control or YAP siRNA and stimulated with IL4/IL13. Data are shown as the mean ± SEM, *n* = 3 for each experimental group. Gene expression data were normalized with the reference gene *Gapdh*, and results are represented as fold change relative to the control treatment. For numerical raw data, please see S1 Data. Arg1, Arginase-I; BMDMs, bone marrow–derived macrophages; Cd206, cluster of differentiation 206; Egr2, early growth response 2; Fizz1, found in inflammatory zone 1; Fpr2, formyl peptide receptor 2; Gapdh, glyceraldehyde-3-phosphate dehydrogenase; IFNγ, interferon gamma; IL, interleukin; LPS, lipopolysaccharide; Nos2; nitric oxide synthase 2; NS, non-significant; PMs, peritoneal macrophages; pYAP, phosphorylated YAP; qRT-PCR, real-time quantitative reverse transcription PCR; siRNA, short interfering RNA; TAZ, transcriptional coactivator with PDZ-binding motif; YAP, yes-associated protein.

the *Tgfβ* and *Vegf* gene expression levels. No significant change in the TGFβ protein level was detected (S2A and S2B Fig). Similarly, to determine whether IL4/IL13 (reparative macrophage-stimuli) treatment resulted in the formation of the reparative and not pro-inflammatory phenotype, we measured the expression of pro-inflammatory marker IL6 and observed no change in the protein levels (S2C Fig). Collectively, pro-inflammatory macrophage-stimuli strongly enhanced both YAP and TAZ protein levels, suggesting a role for YAP/TAZ in macrophage polarization.

To better understand the role of YAP and TAZ in macrophage polarization, we transfected short interfering RNA (siRNA) to knockdown mouse *YAP* and *TAZ* in wild-type BMDMs. The knockdown efficiency of *YAP* and *TAZ* in BMDMs was confirmed by western blot analysis (Fig 1C and S3A Fig). Compared with BMDMs transfected with control siRNA, *YAP* and *TAZ* knockdown BMDMs had reduced levels of pro-inflammatory markers such as IL6 and nitric oxide synthase 2 (Nos2), after LPS/IFNγ treatment (Fig 1C and S3A Fig). Consistent with changes in protein levels, mRNA levels of pro-inflammatory genes such as *IL6*, *IL1β*, *IL12β*, and *Nos*2 were also decreased in *YAP* knockdown BMDMs when compared with controls (Fig 1D). Likewise, compared to control siRNA, *TAZ* knockdown in BMDMs significantly reduced the mRNA levels of *IL1β*, *IL12β*, and *Nos2* (S3B Fig). Similar changes in *IL6*, *IL1β*, and *Nos2* expressions were observed when wild-type BMDMs were treated with Hippo signaling inhibitor verteporfin (VP) in the presence of LPS/IFNγ (S3D Fig). In contrast, knockdown of *YAP* in the presence of IL4/IL13 resulted in significant up-regulation of anti-inflammatory/reparative macrophage marker genes such as *Arg1* and early growth response 2 (*Egr2*). However, no significant change in cluster of differentiation 206 (*Cd206*), *Ym1*, and found in inflammatory zone 1 (*Fizz1*) expression was observed (Fig 1E). Interestingly, knockdown of *TAZ* in the presence of IL4/IL13 increased the mRNA level of *Arg1*, *Ym1*, and *Fizz1* (S3C Fig). Together, these results suggest that YAP and TAZ play an important role in macrophage polarization.

## Global changes in gene expression due to *YAP/TAZ* deletion in BMDMs

To further establish the role of YAP/TAZ in macrophage polarization, we generated myeloid cell-specific *YAP/TAZ* double knockout mice by crossing *YAP^flox/flox;TAZ^flox/flox* mice with lysozyme-cre (*LysM^cre*) mice that drive Cre recombinase activity in myeloid lineages, including macrophages. *YAP^flox/flox;TAZ^flox/flox* and *LysM^cre;YAP^flox/flox;TAZ^flox/flox* mice are referred to as control and *YAP/TAZ* double knockout (*YAP/TAZ*-dKO), respectively. To determine the deletion efficiency, we isolated BMDMs from control and *YAP/TAZ*-dKO mice and performed real-time quantitative reverse transcription PCR (qRT-PCR) and western blot for YAP and TAZ. YAP and TAZ expression both at RNA and protein levels were significantly reduced in

*YAP/TAZ*-dKO BMDMs, suggesting efficient deletion (S4 Fig). To determine the purity of isolated BMDMs and PMs, we performed flow cytometry using cluster of differentiation 11B-fluorescein isothiocyanate (CD11b-FITC) and Allophycocyanin (APC) anti-mouse F4/80 antibodies. We observed nearly 99.69% and 97.63% purity for BMDMs and PMs, respectively (S5A and S5B Fig). Considering YAP/TAZ regulates organ size by modulating cell proliferation, we performed Ki67 immunostaining on BMDMs isolated from control and *YAP/TAZ*-dKO mice. Consistent with previous reports, we did not observe any significant change in macrophage proliferation (S6A and S6B Fig) [45,46]. Similarly, macrophage migration was not affected in both control and *YAP/TAZ*-dKO BMDMs (S6C and S6D Fig).

To identify downstream targets of YAP/TAZ in macrophage polarization, we performed RNA sequencing (RNA-seq) analysis on BMDMs isolated from 2- to 3-month-old control and *YAP/TAZ*-dKO mice and treated them with or without LPS/IFNγ (Fig 2). The average RNA-seq depth was 30 million reads per sample. Paired-end RNA-seq reads from each sample were aligned to the mouse reference genome with an average mapping rate of 81%. Genes with absolute log fold change >1.0 and false discovery rate <5% were considered as differentially expressed. We analyzed the RNA-seq data by comparing control untreated to LPS/IFNγ treated BMDMs, *YAP/TAZ*-dKO untreated to LPS/IFNγ treated BMDMs, and control to *YAP/TAZ*-dKO BMDMs, both treated with LPS/IFNγ. We identified 3,253, 3,191, and 137 differentially expressed genes, respectively (Fig 2A and 2B, S1 FCS file). We also identified 86 genes that were differentially expressed and were common among the 3 groups (Fig 2A). Heat map of top 50 differential genes between LPS/IFNγ treated control to *YAP/TAZ*-dKO groups showed decreased expression of many pro-inflammatory genes such as *IL6*, *Il12b*, C-C chemokine receptor type 7 (*Ccr7*), etc. (Fig 2B). Pathway enrichment analysis identified significant enrichment of genes involved in regulating inflammation such as cellular response to cytokine stimulus, inflammatory response, and cytokine-mediated signaling (Fig 2C). MA plots showed that immune, especially pro-inflammatory macrophage marker genes such as *IL6*, *IL1β*, and *IL12β* were significantly down-regulated in *YAP/TAZ*-dKO cells compared with controls (Fig 2D and S7 Fig).

### *YAP/TAZ* deficiency impairs pro-inflammatory macrophage phenotype

RNA-seq results showed significant changes in immune gene expression after *YAP/TAZ* deletion in the macrophages. To further characterize the pro-inflammatory macrophage polarization defects in *YAP/TAZ*-dKO, we isolated BMDMs from control and *YAP/TAZ*-dKO mice and treated them with/without LPS/IFNγ for 12 hours. Total RNA was then isolated after for qPCR analysis. In unstimulated conditions, *YAP/TAZ*-dKO BMDMs exhibited reduced expression of pro-inflammatory genes (*IL1β* and *IL12β)* (Fig 2E). Similarly, in the presence of LPS/IFNγ, expression of pro-inflammatory genes such as *IL6*, *IL1β*, *IL12β*, *Nos2*, *TNFα*, C–C motif chemokine ligand 2 (*Ccl2*), and *Rantes* was significantly decreased in *YAP/TAZ*-dKO BMDMs (Fig 2E). Additionally, *YAP/TAZ*-deficient BMDMs showed a markedly reduced level of NO after LPS/IFNγ treatment, suggesting a reduced inflammatory response (Fig 2F). To determine whether the changes in gene expression are translated to the protein level, we performed western blot for IL6, IL1β, and Nos2. We observed reduced IL6, IL1β, and Nos2 protein levels in *YAP/TAZ*-dKO BMDMs, when compared with control BMDMs after LPS/IFNγ treatment (Fig 3A). To determine the molecular mechanism by which YAP/TAZ may affect the expression of IL6, we assessed the activation status of key components of the Toll-like receptor 4 (TLR4)-transforming growth factor beta-activated kinase 1 (Tak1)-nuclear factor kappa B (NF-kB)/mitogen-activated protein (MAP) kinases pathway. Upon activation, TLR4 recruit adaptor protein myeloid differentiation primary response 88 (MyD88) and

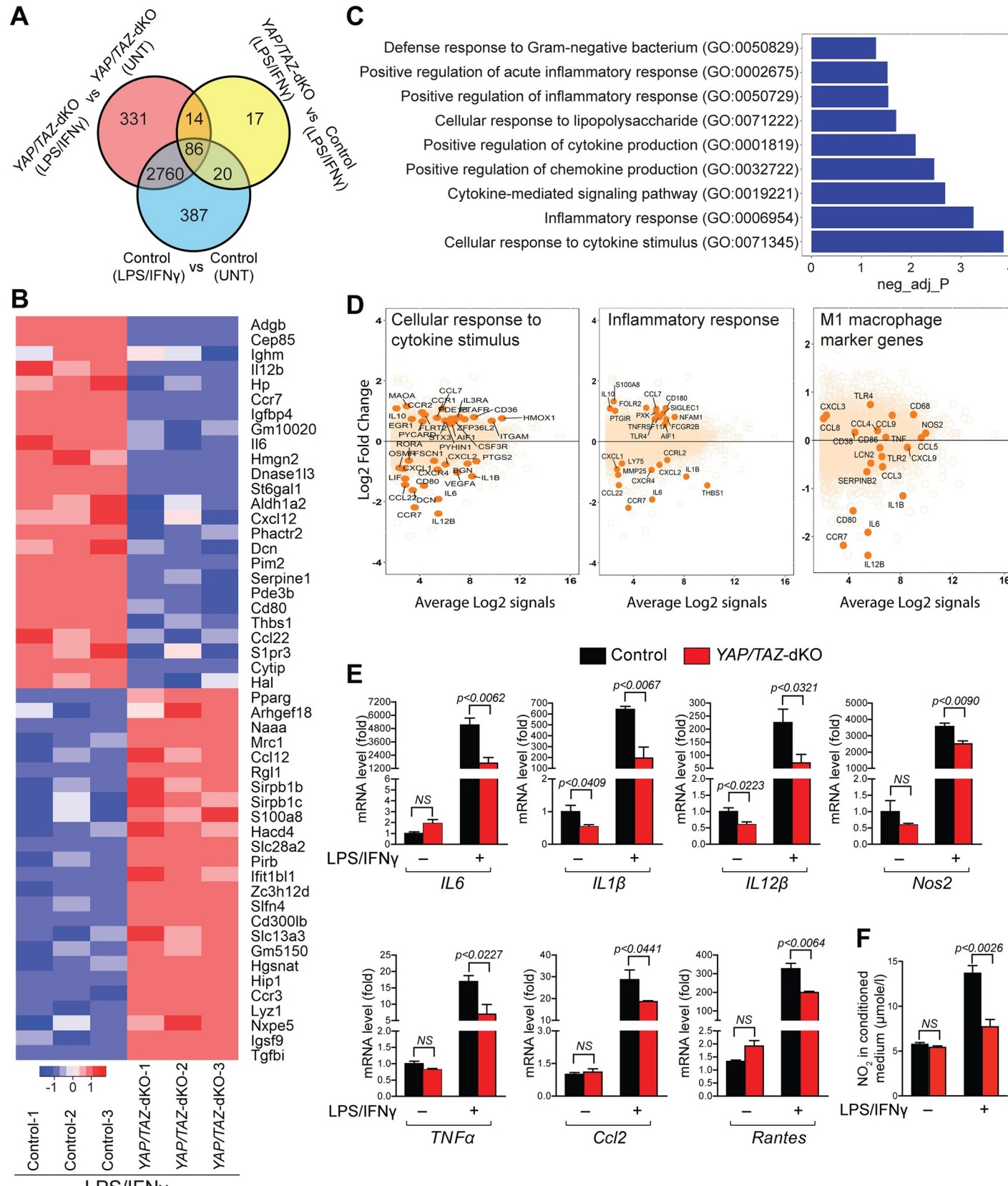

**Fig 2. Macrophage-specific gene expression changes due to *YAP/TAZ* deletion.** (A) Venn analysis of RNA-seq data showing overlap between significantly differentially expressed genes (false discovery rate [FDR] ≤ 5%, absolute log2FC ≥1.0) between control and *YAP/TAZ*-dKO BMDMs with and without LPS/IFNγ treatment. (B) Heat map of 45 differentially expressed genes (out of a total of 86 genes) identified by Venn analysis of RNA-seq data. (C) Pathway enrichment analysis of the RNA-seq data from untreated or LPS/IFNγ treated control and *YAP/TAZ*-dKO BMDMs. (D) MA plots for selected pathways showing top differentially expressed pathway genes. MA plots for pro-inflammatory macrophage marker genes are also presented. (E) BMDMs were isolated from control and *YAP/TAZ*-dKO mice and stimulated with/without LPS/IFNγ for 12 hours. (A) qRT-PCR for pro-inflammatory marker genes, *IL6*, *IL1β*, *IL12β*, *Nos2*, *TNFα*, *Ccl2*, and *Rantes* using RNA isolated from untreated or LPS/IFNγ-treated control and *YAP/TAZ*-dKO BMDMs. *n* = 3 in each group. (F) NO production determined by nitrite (NO₂−) levels in conditioned medium prepared from untreated or LPS/IFNγ-treated control and *YAP/TAZ*-dKO BMDMs. *n* = 3 in each group. For numerical raw data, please see S1 Data. BMDMs, bone marrow–derived macrophages; Ccl2, C–C motif chemokine ligand 2; FDR, false discovery rate; IFNγ, interferon gamma; IL, interleukin; LPS, lipopolysaccharide; NO, nitric oxide; Nos2, nitric oxide synthase 2; qRT-PCR, real-time quantitative reverse transcription PCR; RNA-seq, RNA sequencing; TAZ, transcriptional coactivator with PDZ-binding motif; TNFα, tumor necrosis factor alpha; UNT, untreated; YAP, yes-associated protein; YAP/TAZ-dKO, YAP/TAZ double knockout.

activate common upstream activator Tak1 of NF-kB and MAP kinase pathway [47]. We found that there was a significant reduction of phosphorylated Tak1 protein level in *YAP/TAZ*-dKO BMDMs after 15 minutes of LPS/IFNγ treatment (Fig 3B). However, we did not observe any changes in the phosphorylated level of inhibitor of nuclear factor kappa-αβ-kinase (IKKαβ) (Fig 3B). Consistently, YAP did not affect the activity of *NF-κB* luciferase reporter in unstimulated condition suggesting that YAP/TAZ mediated activation of IL6 in macrophage maybe not dependent on the NF-kB signaling pathway (S8A Fig). Next, we tested the key components of MAPKs and observed significant but transient reduction in the levels of phosphorylated c-Jun N-terminal kinase (JNK), p38, and extracellular signal-regulated kinase 1/2 (Erk1/2) in *YAP/TAZ*-deficient macrophages after 15 minutes of LPS/IFNγ stimulation (Fig 3B).

Considering YAP and TAZ are transcriptional coactivators and interact with Tead (Transcriptional enhanced associate domain) transcription factors to regulate target gene expression, we next assessed whether YAP/TAZ regulate *IL6* expression at the transcription level by directly binding and modulating *IL6* promoter activity. We analyzed the promoter of *IL6* and identified 10 consensus Tead-binding sequences (TBSs) within the 2Kb promoter fragment (Fig 3C and 3D). We then PCR-amplified the *IL6* promoter fragment (2kb), cloned into a luciferase reporter plasmid, and tested in luciferase reporter assays. Both YAP and TAZ strongly activated the *IL6* promoter-luciferase activity (Fig 3E). Chromatin immunoprecipitation (ChIP) assays demonstrated direct binding of YAP and TAZ to *IL6* promoter (Fig 3F). Mutation of TBS sites within the *IL6* promoter fragments significantly reduced YAP's ability to activate these promoter fragments in the luciferase reporter assays (Fig 3G). To further demonstrate that Tead binding is necessary for YAP-mediated activation of *IL6* promoter, Hippo signaling inhibitor VP, was added 24 hours after transfection. Activation of *IL6* promoter by YAP or TAZ was completely abolished by VP (Fig 3H). We next performed luciferase assay using IL6 reporter co-transfected with YAP plasmid in the presence or absence of VP or MAPK inhibitor SB203580 (S8B and S8C Fig). We observed a mild reduction in the luciferase activity in the presence of SB203580. In contrast, VP completely blocked the YAP-medicated IL6 activation suggesting that direct regulation of IL6 by YAP may play a dominant role compared to the MAPK pathway in mediating IL6 activation (S8B and S8C Fig). Together, these results demonstrate that YAP/TAZ promote pro-inflammatory macrophage phenotype by modulating IL6 production.

### *YAP/TAZ* deficiency promotes reparative macrophage phenotype

To characterize the reparative macrophage phenotype in *YAP/TAZ*-dKO, we isolated BMDMs from control and *YAP/TAZ*-dKO mice and treated them with/without IL4/IL13 for 12 hours. Total RNA was isolated for qPCR analysis. In both unstimulated and stimulated conditions, *YAP/TAZ*-dKO BMDMs exhibited increased levels of reparative genes such as *Arg1*, *Egr2*,

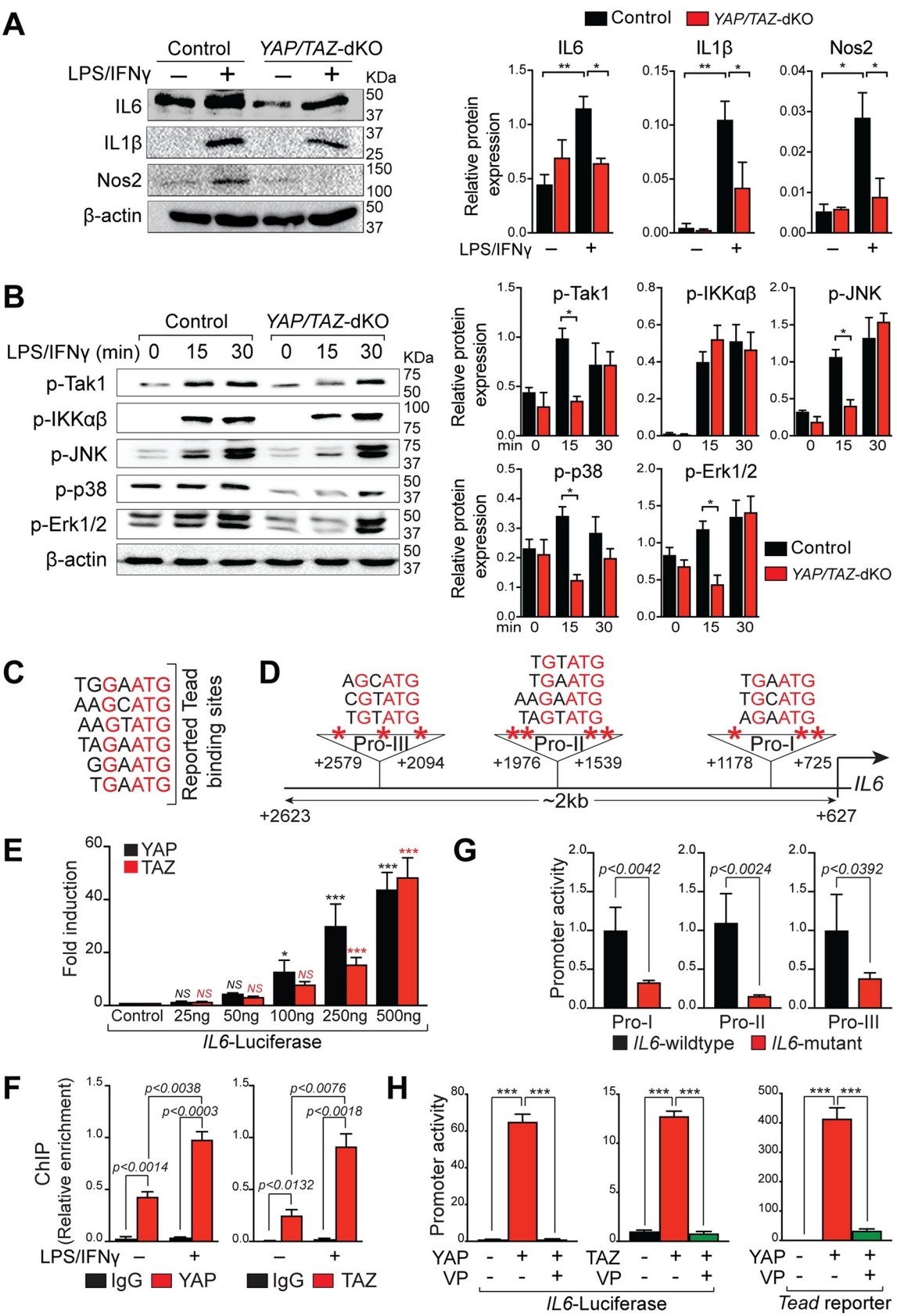

**Fig 3. *YAP/TAZ* deficiency impairs pro-inflammatory macrophage phenotype.** (A) BMDMs were isolated from control and *YAP/TAZ*-dKO mice and stimulated with/without LPS/IFNγ for 12 hours. Western blot analysis for IL6, IL1β, and Nos2 was performed using total lysates from untreated or LPS/IFNγ-treated control and *YAP/TAZ*-dKO BMDMs. β-actin is shown as a loading control. The relative protein expression was quantified. (B) BMDMs were isolated from control and *YAP/TAZ*-dKO mice and stimulated with/without LPS/IFNγ for 15 and 30 minutes. The phosphorylated level of JNK, p38, Erk1/2, IKKαβ, and Tak1 was detected by western blot analysis, and the relative expression was quantified. β-actin is shown as a loading control. (C) Reported TBSs. (D) Predicted TBSs in *IL6* promoter. *IL6*-promoter fragments I (Pro-I) and III (Pro-III) harbor 3 TBSs, while fragment II (Pro-II) harbors 4 TBSs. (E) Results of normalized luciferase reporter assays in HEK293T cells with full-length *IL6*-luciferase reporter (approximately 2Kb promoter) in the presence of YAP or TAZ. (F) ChIP assay using chromatin from wild-type untreated or LPS/IFNγ treated BMDMs with IgG, YAP, and TAZ antibody. (G) *IL6*-promoter fragments (Pro-I, Pro-II, and Pro-III) containing wild-type TBSs driving luciferase were compared to mutant fragments with mutated TBSs. (H) *IL6* and *Tead*-luciferase reporters were transfected in HEK293T cells with/without YAP or TAZ in the presence or absence of verteporfin (5 μM). Data from luciferase experiments are presented as mean ± SD. For numerical raw data, please see S1 Data. BMDMs, bone marrow–derived macrophages; ChIP, chromatin immunoprecipitation; Erk1/2, extracellular signal-regulated kinase 1/2; HEK293T, human embryonic kidney 293 T; IFNγ, interferon gamma; IgG, immunoglobulin G; IKKαβ, inhibitor of nuclear factor kappa-αβ-kinase; IL, interleukin; JNK, c-Jun N-terminal kinase; LPS, lipopolysaccharide; Nos2, nitric oxide synthase 2; SD, standard deviation; Tak1, transforming growth factor beta-activated kinase 1; TAZ, transcriptional coactivator with PDZ-binding motif; TBSs, Tead-binding sequences; Tead, Transcriptional enhanced associate domain; VP, verteporfin; YAP, yes-associated protein; YAP/TAZ-dKO, YAP/TAZ double knockout.

*Cd206*, *Ym1*, and fibronectin 1 (*Fn1*) (Fig 4A). Given that reparative (M2) macrophages can be subdivided into 3 categories—namely, M2a, M2b, and M2c. We, therefore, stimulated control and *YAP/TAZ*-dKO BMDMs with respective stimuli and analyzed the marker gene expressions for all 3 reparative subcategories. Consistently, the expression of reparative subset genes was also increased in *YAP/TAZ*-dKO BMDMs. For example, the expression of marker genes for M2a (*Cd206* and *Ym1*), M2b (*Il10* and *Il1ra*), and M2c (matrix metallopeptidase 9 (*Mmp9*)) was increased in *YAP/TAZ*-dKO BMDMs (S9A–S9C Fig). Alternatively, a decrease in the expression of the M2c marker gene *Tgfβ* was observed in *YAP/TAZ*-dKO BMDMs compared to control under M2c-stimulated conditions (S9C Fig). Gene expression data were further supported by western blot analysis for Arg1, showing increased protein expression in *YAP/TAZ*-dKO BMDMs (Fig 4B). Activated PI3K/AKT pathway has also been recognized as an essential step toward reparative macrophage polarization as its inhibition abrogates the up-regulation of reparative genes [48]. We, therefore, analyzed the levels of phosphorylated AKT in control and *YAP/TAZ*-dKO BMDMs treated with IL4/IL13. We observed a significant increase in phosphorylated AKT level in *YAP/TAZ*-deficient macrophages after 30 minutes of IL4/IL13 treatment suggesting that *YAP/TAZ* deficiency promotes reparative phenotype (Fig 4C). A recent study has demonstrated that p53 is expressed in both pro-inflammatory and reparative macrophages and regulates reparative phenotype [49]. Surprisingly, our RNA-seq analysis did not detect any change in tumor protein 53 (*Trp53*) expression during pro-inflammatory macrophage polarization (S10A Fig). Similarly, p53 levels were not altered during reparative macrophage polarization in *YAP/TAZ*-deficient BMDMs suggesting that YAP/TAZ-mediated macrophage polarization is not dependent on p53 (S10B Fig).

Next, to further examine how YAP/TAZ are involved in reparative macrophage polarization, we treated RAW264.7 cells, PMs, and BMDMs with IL4/IL13 for 0 to 24 hours and performed YAP, pYAP, and TAZ western blot analysis. We observed increased expression of YAP after 6 to 8 hours of stimulation and remained steady for 12 to 24 hours, in all tested cell types (Fig 4D–4F). Similarly, the phosphorylated level of YAP was also elevated. Similar to YAP, the expression of TAZ was also enhanced in all 3 macrophage cells upon IL4/IL13 treatment (Fig 4D–4F). Moreover, since YAP/TAZ exert their functions in the nucleus, we furthermore measured the nuclear abundance of YAP/TAZ together with cytoplasmic expression of pYAP in BMDMs after IL4/IL13 treatment for 8 and 12 hours. Western blot analysis revealed a significant increase of YAP, TAZ, and pYAP protein levels upon IL4/IL13 treatment (S1 Fig).

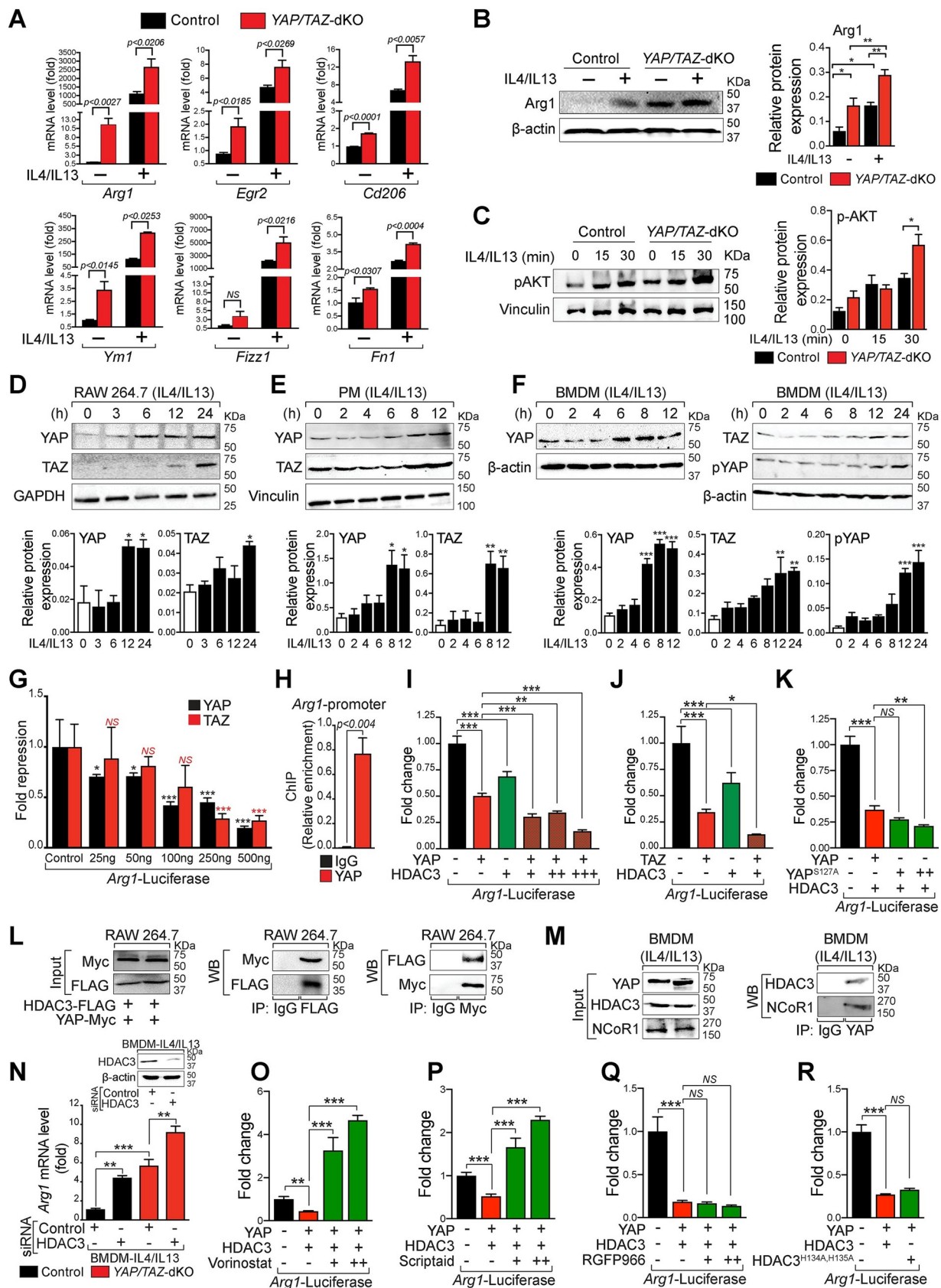

**Fig 4. YAP interacts with the HDAC3-NCoR1 repressor complex to inhibit reparative macrophage phenotype.** (A) BMDMs were isolated from control and *YAP/TAZ*-dKO mice and stimulated with/without IL4/IL13 for 12 hours. qRT-PCR for reparative marker genes, *Arg1*, *Egr2*, *Cd206*, *Ym1*, *Fizz1*, *and Fn1* using RNA isolated from untreated or IL4/IL13-treated control and *YAP/TAZ*-dKO BMDMs. *n* = 3 in each group. Gene expression results were normalized to *gapdh*, and results are represented as fold change. (B) Western blot analysis for Arg1 was performed using total lysates from untreated or IL4/IL13-treated control and *YAP/TAZ*-dKO BMDMs. β-actin is shown as a loading control. The relative protein expression was quantified. (C) BMDMs were isolated from control and *YAP/TAZ*-dKO mice and stimulated with/without IL4/IL13 for 15 and 30 minutes. The phosphorylated level of AKT was detected by WB analysis, and the relative protein expression was quantified. Vinculin is shown as a loading control. (D–F) WB analysis for YAP, pYAP (S127), and TAZ was performed using total lysates from RAW264.7 cells, wild-type PMs, and BMDMs treated with IL4/IL13 for 0 to 24 hours as indicated. β-actin, GAPDH, or vinculin are shown as a loading control. The relative protein expression was quantified. (G) Results of normalized luciferase reporter assays in HEK293T cells with *Arg1*-luciferase reporter in the presence of YAP or TAZ. (H) ChIP assay using chromatin from wild-type IL4/IL13 treated BMDMs with IgG or YAP antibody. (I–K) Results of normalized luciferase reporter assays in HEK293T cells with *Arg1*-luciferase reporter in the presence of YAP, TAZ, YAP$^{S127A}$, and HDAC3 alone or as indicated combinations. (L) RAW264.7 cells were co-transfected with YAP-Myc and HDAC3-FLAG plasmids for 48 hours and then treated with IL4/IL13 for 12 hours. IP was performed using IgG control, anti-FLAG, and anti-Myc antibodies followed by WB for the Myc or FLAG tag. (M) Wild-type BMDMs were treated with IL4/IL13 for 12 hours, and IP was performed using IgG or an anti-YAP antibody followed by WB for YAP, HDAC3, and NCoR1. (N) BMDMs were isolated from control and *YAP/TAZ*-dKO mice and transfected with control, YAP, or HDAC3 siRNA for 72 hours, followed by IL4/IL13 stimulation for 16 hours. RNA was prepared for qRT-PCR analysis of *Arg1*. (O–Q) Results of normalized luciferase reporter assays in HEK293T cells with *Arg1*-luciferase reporter when YAP, HDAC3 alone, or their combinations were transfected in the presence or absence of vorinostat, scriptaid, or RGFP966. (R) Results of normalized luciferase reporter assays in HEK293T cells transfected with *Arg1*-luciferase reporter with or without YAP. After 24 hours, cells were infected with either HDAC3 or HDAC3$^{H134A,\ H135A}$ lentivirus and analyzed 48 hours after infection. Data from luciferase experiments are presented as mean ± SD. For numerical raw data, please see S1 Data. Arg1, Arginase-I; BMDMs, bone marrow–derived macrophages; Cd206, cluster of differentiation 206; Egr2, early growth response 2; Fizz1, found in inflammatory zone 1; Fn1, fibronectin 1; GAPDH, glyceraldehyde-3-phosphate dehydrogenase; HDAC3, histone deacetylase 3; HEK293T, human embryonic kidney 293 T; IgG, immunoglobulin G; IL, interleukin; IP, immunoprecipitation; NCoR1, nuclear receptor corepressor 1; NS, non-significant; PMs, peritoneal macrophages; pYAP, phosphorylated YAP; qRT-PCR, real-time quantitative reverse transcription PCR; siRNA, short interfering RNA; WB, western blot; YAP, yes-associated protein; YAP/TAZ-dKO, YAP/TAZ double knockout.

To determine how YAP/TAZ inhibit *Arg1* expression, we analyzed the promoter of *Arg1* and identified 10 consensus TBSs within the 3.1Kb promoter fragment and first exon (S11 Fig). We then tested the *Arg1* promoter fragment in luciferase reporter assays. Both YAP and TAZ strongly inhibited the *Arg1* promoter-luciferase activity in a dose-dependent manner (Fig 4G). ChIP assays demonstrated direct binding of YAP to the *Arg1* promoter (Fig 4H). To determine the molecular mechanism of YAP/TAZ-mediated transcriptional repression further, we tested the hypothesis that YAP/TAZ interact with HDAC3-NCoR1 complex to repress *Arg1* expression in macrophages. We observed that *Arg1* promoter-luciferase activity was further repressed when both YAP and HDAC3 were co-expressed compared to YAP alone (Fig 4I). A similar repressive response to *Arg1* promoter-luciferase activity was obtained when TAZ was co-expressed together with HDAC3 (Fig 4J). The repressive response to *Arg1* promoter-luciferase activity was further enhanced when HDAC3 was co-expressed together with YAP$^{S127A}$ (a constitutively active form that remains in the nucleus and is transcriptionally active) compared to YAP (Fig 4K).

To determine whether YAP physically interacts with HDAC3-NCoR1 repressor complexes, we co-transfected RAW264.7 cells with plasmid constructs expressing YAP-Myc and HDAC3-FLAG, followed by immunoprecipitation using either FLAG-tag or Myc-tag followed by western blot for Myc and FLAG respectively. We found that YAP interacted with HDAC3 in RAW264.7 cells (Fig 4L). We extended these studies further and treated wild-type BMDMs with IL4/IL13 and performed immunoprecipitation experiments using antibody against YAP and western blot for HDAC3 and NCoR1. We detected both HDAC3 and NCoR1 protein in the YAP immunoprecipitated samples compared to immunoglobulin G (IgG) controls suggesting an interaction between YAP, HDAC3, and NCoR1 (Fig 4M). To determine if HDAC3 exerts its repressive function exclusively by interacting with YAP/TAZ on *Arg1* promoter, we performed a knockdown experiment using *HDAC3* siRNA followed by IL4/IL13 treatment on BMDMs from control and *YAP/TAZ*-dKO mice. The knockdown efficiency of *HDAC3* in BMDMs was confirmed by western blot analysis (Fig 4N). In control BMDMs, *HDAC3*

knockdown resulted in the up-regulation of *Arg1* expression (approximately 5-fold) compared to control siRNA. In *YAP/TAZ*-dKO BMDMs, *Arg1* expression was elevated; however, the knocking down of *HDAC3* further increased (approximately 2-fold) the *Arg1* expression suggesting that HDAC3 can also modulate *Arg1* expression by alternative mechanisms (Fig 4N).

To further understand the structural versus enzymatic (deacetylase activity) role of HDAC3 in the repression of Arg1, we utilized pan-HDAC inhibitor vorinostat and scriptaid as well as HDAC3-specific inhibitor RGFP966. First, we co-transfected the HEK293T cells with both YAP and HDAC3 constructs in the presence or absence of vorinostat or scriptaid. We observed that both vorinostat or scriptaid treatment abolished the YAP-HDAC3 mediated repression of *Arg1* (Fig 4O and 4P). This was not surprising, as both vorinostat and scriptaid are known to affect both protein expression as well as the deacetylase activity of HDACs including HDAC3 [50–52]. To further support our findings that the enzymatic function of HDAC3 is not essential for HDAC3-mediated repression of *Arg1*, we utilized RGFP966, an HDAC3-specific inhibitor. In multiple biological systems, RGFP966 has been shown to affect the deacetylase activity but not the protein expression of HDAC3 [53,54]. We observed that RGFP966 treatment did not affect the YAP-HDAC3 mediated repression of *Arg1* (Fig 4Q). To further demonstrate that the deacetylase activity of HDAC3 is not required for *Arg1* repression, we utilized a previously described mutant form of HDAC3 in which 2 highly conserved tandem His residues, 134 and 135, are mutated to alanine (HDAC3$^{H134A, H135A}$). These mutations do not affect its expression, chromatin recruitment, and interaction with NCOR1; however, they make HDAC3 completely enzymatically inactive [55,56]. We observed that HDAC3$^{H134A, H135A}$ mutant was able to repress the Arg1 reporter similar to HDAC3 suggesting that the deacetylase activity of HDAC3 is not required for *Arg1* repression (Fig 4R).

## YAP/TAZ negatively regulates the bactericidal activity of macrophages

Since *YAP/TAZ*-deficient macrophages display reduced pro-inflammatory phenotype, we hypothesized that they may have reduced bactericidal activity. We performed the bacterial killing assay, in which BMDMs from control and *YAP/TAZ*-dKO mice were infected with either gram-negative (*Escherichia coli*) or gram-positive (*Listeria monocytogenes*) bacteria for 2 hours. The BMDMs were then thoroughly washed and incubated for 16 hours (S12A Fig). *YAP/TAZ*-deficient macrophages showed significantly decreased bactericidal activity toward both gram-negative (S12B and S12C Fig) and gram-positive bacteria (S12F and S12G Fig), as evident by increased bacterial colony formation. To examine the molecular basis of bacterial killing, we also measured the levels of NO production and consequently performed pro-inflammatory marker gene expression analysis. We found that both NO and expression of *IL6*, *IL1β*, *IL12*, *Rantes*, and *Nos2* were significantly down-regulated in *YAP/TAZ*-deficient macrophages when infected with both gram-negative and gram-positive bacteria (S12D, S12E, S12H and S12I Fig). We did not observe any significant alterations in NO and pro-inflammatory gene expression at the basal level. However, we observed that *Rantes* and *Nos2* expressions were decreased in *YAP/TAZ*-dKO macrophages after *L. monocytogenes* infection (S12D, S12E, S12H and S12I Fig). To further investigate the effect of YAP activation on bacterial killing efficiency, we isolated BMDMs from control and *LysM*$^{Cre}$; *R26*$^{YAP5SA}$ mice and performed similar bacterial killing experiments described above. BMDMs isolated from *LysM*$^{Cre}$; *R26*$^{YAP5SA}$ mice displayed a significantly greater bactericidal activity toward both gram-negative and gram-positive bacteria. Increased bactericidal activity in *LysM*$^{Cre}$; *R26*$^{YAP5SA}$ BMDMs was associated with increased production of NO, as well as higher expression of pro-inflammatory genes such as *IL6*, *IL1β*, *Rantes*, and *Nos2* (S13 Fig).

### YAP/TAZ-deficient mice show improved cardiac function recovery post-MI

Macrophage polarization is a common event after MI, where macrophage phenotypes control the degree of cardiac damage, repair, and function [6,14]. We, therefore, investigated whether *YAP/TAZ* deficiency affects macrophage polarization and cardiac function after MI. We initially examined the cardiac function in *YAP/TAZ*-dKO and control mice at baseline and 4-week post-MI using echocardiographic (echo) analysis. Under baseline conditions, *YAP/TAZ*-dKO hearts show normal left ventricular ejection fraction (LVEF), fractional shortening (FS), left ventricular end-systolic volume (LVESV), left ventricular end-diastolic volume (LVEDV), stroke volume (SV), fractional area change (FAC), and left ventricular mass (LV Mass) similar to their littermate controls (Fig 5A–5E and S14 Fig). However, LVEF in control is severely reduced from an average of 66.34% to 24.94% (62.4% reduction compared to baseline) post-MI. In contrast, we observed significantly preserved LVEF in *YAP/TAZ*-dKO, from 60.24% at baseline to 42.17% post-MI (29.99% reduction compared to baseline) (Fig 5A). FS was not affected at baseline in both control and *YAP/TAZ*-dKO mice. However, post-MI, FS is significantly preserved in *YAP/TAZ*-dKO (from 16.63% at baseline to 9.93% post-MI; 40.28% reduction compared to baseline) compared with controls (from 17.84% at baseline to 5.83% post-MI; 67.32% reduction compared with baseline) (Fig 5B). LVESV was not affected at baseline in both control and *YAP/TAZ*-dKO mice. However, post-MI, LVESV is significantly reduced in *YAP/TAZ*-dKO (from 14.93 μL at baseline to 52 μL post-MI; 3.48 fold more blood retention compared to baseline) compared with controls (from 12.79 μL at baseline to 78.84 μL post-MI; 6.16 fold more blood retention compared to baseline) (Fig 5C). FAC in control is severely reduced from an average of 48.65% to 16.41% (66.26% reduction compared to baseline) post-MI. In contrast, we observed significantly preserved FAC in *YAP/TAZ*-dKO, from 42.83% at baseline to 25.49% post-MI (40.48% reduction compared to baseline) (Fig 5D). In addition, the SV was also improved from 24.73% in controls to 34.81% (40.47% increase) in *YAP/TAZ*-dKO mice compared to controls after MI. No significant change in the LVEDV and LV mass was observed (S14 Fig).

Next, we investigated whether protected cardiac function in *YAP/TAZ*-deficient mice after MI is due to changes in macrophage subpopulation and pro-inflammatory and reparative gene expression. We quantified the macrophage numbers by flow cytometry and examined the expression of pro-inflammatory and reparative genes from isolated macrophages from control and *YAP/TAZ*-dKO mice after MI. At day 2 post-MI, the number of pro-inflammatory macrophages (F4/80$^+$iNOS$^+$) was significantly reduced in *YAP/TAZ*-deficient hearts compared with control hearts (Fig 5F, S5C Fig, S20 Fig and S1–S9 FCS files). However, at day 6 post-MI, the percentage of reparative macrophages (F4/80$^+$CD206$^+$) was increased in *YAP/TAZ*-deficient hearts relative to controls (Fig 5F, S5D Fig, S21 Fig and S10–S22 FCS files). The expression profiling for pro-inflammatory and reparative genes reflected the similar results in fluorescence-activated cell sorting (FACS)-sorted macrophages, where levels of pro-inflammatory gene such as *IL6*, *IL1β*, *and Nos2* were reduced, while expressions of reparative genes such as *Arg1*, *Ym1*, and *Fizz1* were elevated in *YAP/TAZ*-deficient mice (Fig 5G and 5H). We also performed pro-inflammatory and reparative gene expression analysis on heart tissues (LV) isolated from control and *YAP/TAZ*-dKO mice after 3 and 7 days post-MI. We observed reduced expression of pro-inflammatory genes such as *IL6*, *IL1β*, *TNFα*, *Nos2*, *Ccl2*, *Rantes*, and *Ccr7* in *YAP/TAZ*-dKO hearts compared with controls during early inflammatory phase at day 3 post-MI, indicating impaired pro-inflammatory function (Fig 5I). In contrast, we detected increased expression of reparative genes such as *Arg1*, *Ym1*, and *Egr2* and decreased levels of pro-fibrotic genes such as *IL10* and *TGFβ1* in *YAP/TAZ*-dKO hearts when compared with controls during early reparative phage at day 7 post-MI. No significant change was detected in the expression of other reparative genes such as *Fizz1* and *Cd206* (Fig 5J).

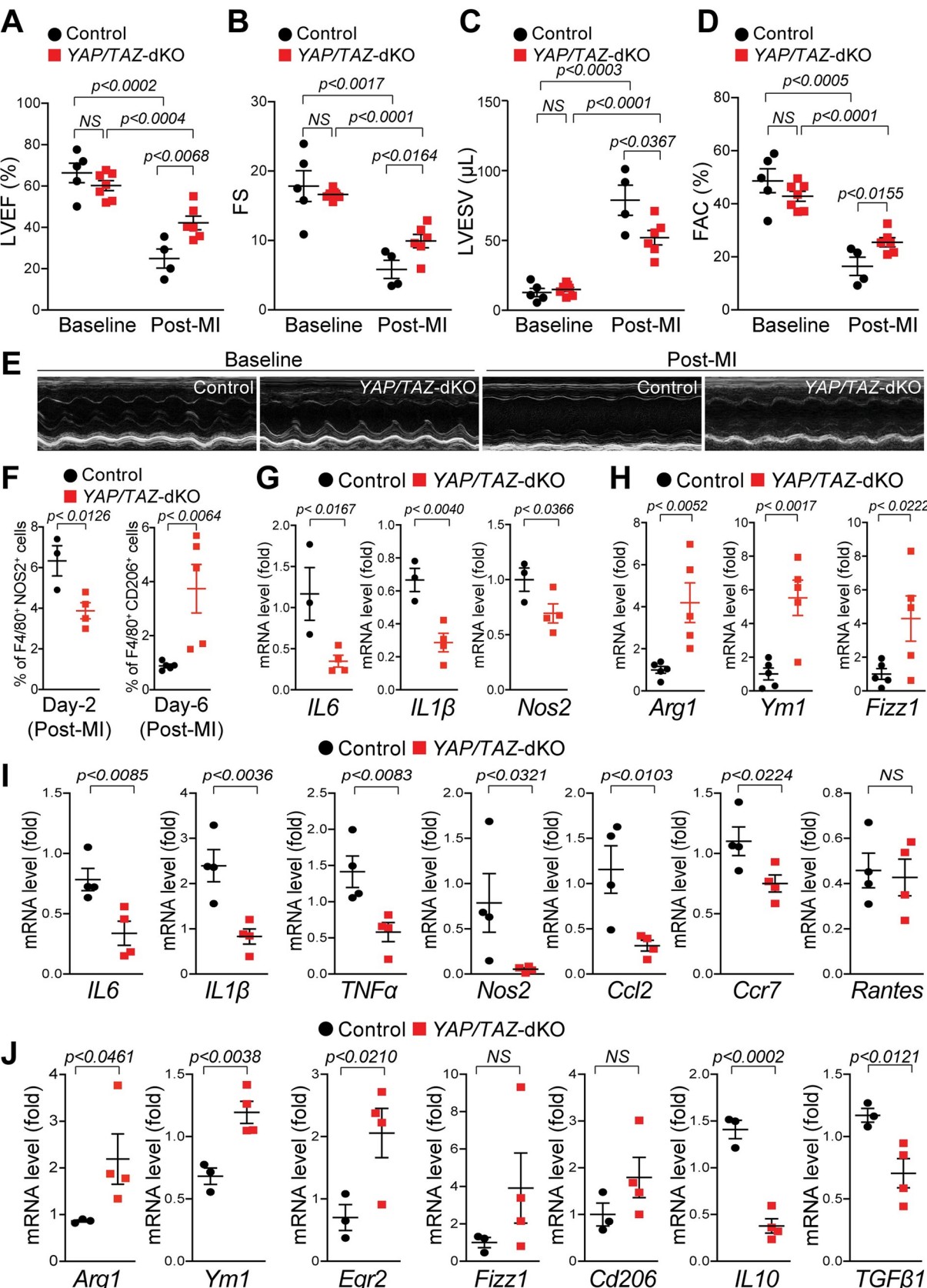

**Fig 5. Improved recovery of cardiac function in *YAP/TAZ*-deficient mice post-MI.** (A–E) Echocardiographic measurements of ventricular functional parameters at baseline and 4-week post-MI from control and *YAP/TAZ*-dKO mice: (A) LVEF; (B) FS; (C) LVESV; (D) FAC; (E) Representative echocardiographic M-mode images of LV. Data are shown as mean ± SEM (*n* = 4 to 7 per group). (F–H) Flow cytometry analysis of pro-inflammatory (iNOS$^+$/F4/80$^+$) and reparative (CD206$^+$/F4/80$^+$) macrophages from the 2 and 6 days post-MI hearts: (F) percentage of pro-inflammatory (F4/80+iNOS+) and reparative (F4/80+CD206+) macrophages. Underlying raw data can be found in S1 Data and S1, S2, S3, S4, S5, S6, S7, S8, S9, S10, S11, S12, S13, S14, S15, S16, S17, S18, S19, S20, S21 and S22 FCS files. (G) qRT-PCR for pro-inflammatory cytokines *IL6*, *IL1β*, and *Nos2* in isolated iNOS$^+$/F4/80$^+$ macrophages from the 2 days post-MI hearts. (H) qRT-PCR for anti-inflammatory markers *Arg1*, *Ym1*, and *Fizz1* in isolated CD206$^+$/F4/80$^+$ macrophages from the 6 days post-MI hearts. (I) qRT-PCR for pro-inflammatory marker genes, *IL6*, *IL1β*, *TNFα*, *Nos2*, *Ccl2*, *Ccr7*, and *Rantes* using RNA isolated from control and *YAP/TAZ*-dKO hearts (LV) at 3 days post-MI. *n* = 4 in each group. (J) qRT-PCR for anti-inflammatory marker genes, *Ym1*, *Egr2*, *Arg1*, *Fizz1*, *Cd206*, *IL10*, and *TGFβ* using RNA isolated from control and *YAP/TAZ*-dKO hearts (LV) at 7 days post-MI. *n* = 3 to 4 in each group. Gene expression results were normalized to *Gapdh*, and results are represented as fold change. For numerical raw data, please see S1 Data. Arg1, Arginase-I; Ccl2, C–C motif chemokine ligand 2; Ccr7, C-C chemokine receptor type 7; Cd206, cluster of differentiation 206; Egr2, early growth response 2; FAC, fractional area change; Fizz1, found in inflammatory zone 1; FS, fractional shortening; Gapdh, glyceraldehyde-3-phosphate dehydrogenase; IL, interleukin; LVEF, left ventricular ejection fraction; LVESV, left ventricular end-systolic volume; MI, myocardial infarction; Nos2, nitric oxide synthase 2; NS, non-significant; qRT-PCR, real-time quantitative reverse transcription PCR; TAZ, transcriptional coactivator with PDZ-binding motif; TGFβ, transforming growth factor beta; TNFα, tumor necrosis factor alpha; YAP, yes-associated protein; YAP/TAZ-dKO, YAP/TAZ double knockout.

Recruitment of neutrophils to the infarcted area is 1 of the earliest events after MI; we, therefore, investigated whether myeloid-specific *YAP/TAZ* deficiency affects the neutrophil recruitment/accumulation to the injured myocardium. At day 2 post-MI, the number of myeloperoxidase positive cells (MPO$^+$), a well-known neutrophil marker, was not alternated in *YAP/TAZ*-deficient hearts compared with control hearts (S15A Fig). Similarly, in vitro neutrophil migration was also not affected due to *YAP/TAZ* deletion (S15B Fig).

MI injury leads to hypoxia and oxidative stress, which ultimately induces reactive oxygen species production that accelerates the progression of ischemic heart disease.

To identify the factors affecting YAP/TAZ expression in the post-MI heart, we tested the hypothesis that hypoxia may cause up-regulation of YAP/TAZ in macrophages. We treated BMDMs with $H_2O_2$ to mimic hypoxic condition and performed western blot analysis. The protein expression of YAP, TAZ, and Nos2 was significantly elevated after $H_2O_2$ treatment demonstrating that ischemic condition can stimulate YAP/TAZ expression in macrophage (S16 Fig).

## Improved post-MI ventricular remodeling and angiogenesis in *YAP/TAZ*-deficient hearts

*YAP/TAZ*-deficient hearts exhibited decreased pro-inflammatory and increased reparative response following MI. To determine the impact of these changes on cardiac tissue repair, we first quantified the initial ischemic area in both control and *YAP/TAZ*-dKO hearts at days 2 post-MI. No significant change was observed between the 2 groups (S17A Fig). Next, we sought to study the impact of these changes on cardiac tissue repair processes such as fibrosis, hypertrophy, and angiogenesis at days 2, 7, and 28 post-MI. Masson's trichrome staining showed no alteration in fibrosis at day 2 post-MI in *YAP/TAZ*-dKO hearts compared to control (S17B Fig); however, a significantly reduced fibrotic area was detected in *YAP/TAZ*-dKO hearts compared with controls at day 28 post-MI (Fig 6A and 6B). Additionally, interstitial myocardial fibrosis was also attenuated in *YAP/TAZ*-dKO heart at day 28 post-MI compared to controls (S18A and S18B Fig). Consistent with reduced fibrosis, expression of collagen (type I), the most abundant isoform found in fibrotic tissue, was significantly reduced in *YAP/TAZ*-dKO hearts compared with controls (Fig 6C and 6D). We then examined the mRNA level of genes involved in pro-fibrotic events. We observed reduced expression of genes required for myofibroblasts formation (smooth muscle actin alpha 2 (*Acta2*) and transgelin (*Tagln*)), collagen biosynthesis (collagen type I α 1 (*Col1α1*), prolyl 4-hydroxylase subunit alpha-1 (*P4ha1*),

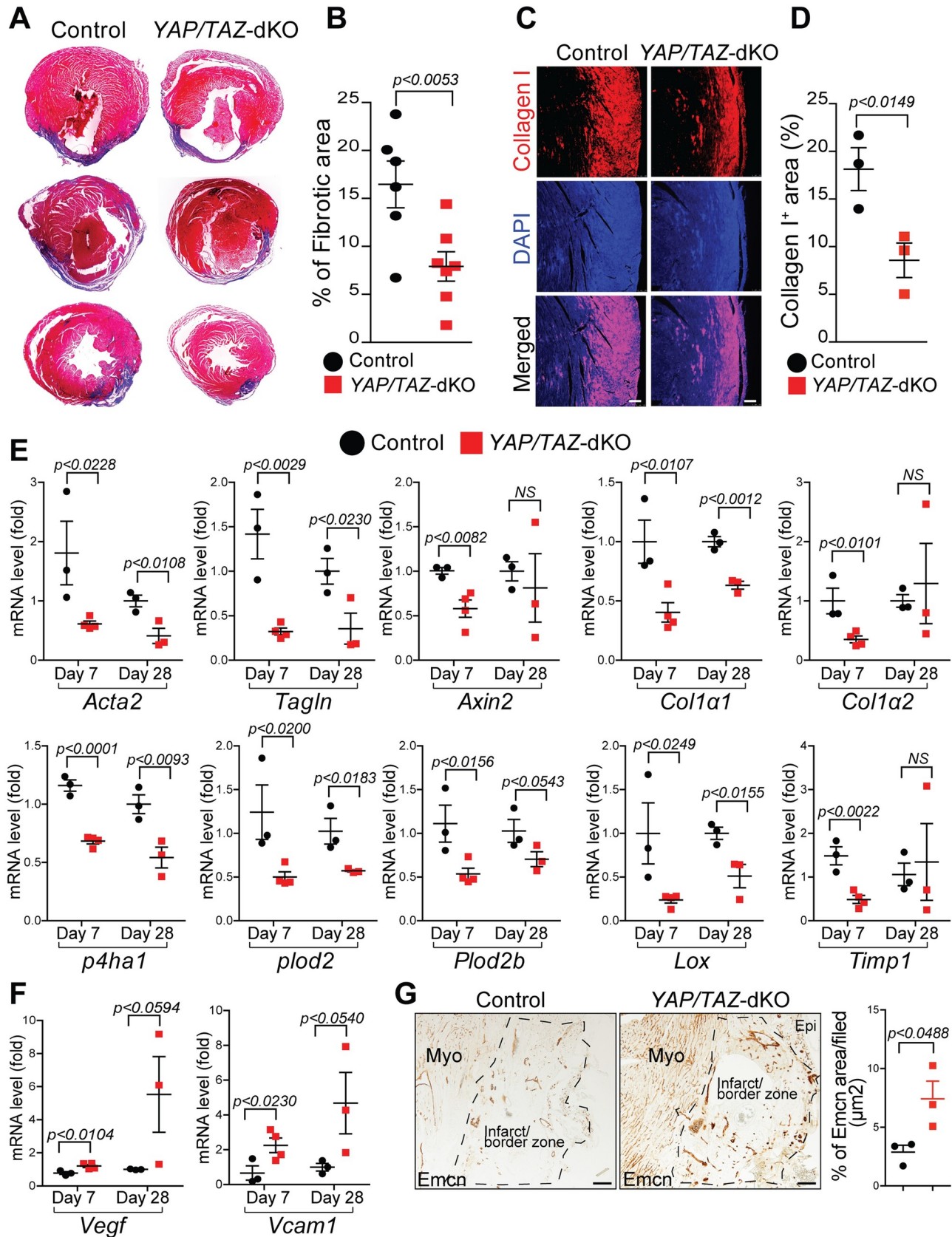

**Fig 6. Improved post-MI ventricular remodeling and angiogenesis in *YAP/TAZ*-deficient hearts.** (A) Masson's trichrome staining on control and *YAP/TAZ*-dKO hearts at 28 days post-MI. (B) Quantification of Masson's trichrome stained fibrotic area in control and *YAP/TAZ*-dKO hearts at 28 days post-MI (*n* = 6 to 7 per group). Fibrotic area was normalized to the remaining heart for each heart section. (C) Immunostaining showing the expression of collagen type I in infarct zone on control and *YAP/TAZ*-dKO heart sections at 28 days post-MI. DAPI was used to stain nuclei. Scale bar 100 μM. (D) Quantification of collagen type I positive area in control and *YAP/TAZ*-dKO heart sections. *n* = 3 per group. (E) qRT-PCR for profibrotic genes involved in collagen biosynthesis using RNA isolated from control and *YAP/TAZ*-dKO hearts (LV) at 7 and 28 days post-MI. *n* = 3 to 4 in each group. (F) qRT-PCR for angiogenic factors *Vegf* and *Vcam1* using RNA isolated from control and *YAP/TAZ*-dKO hearts (LV) at 7 and 28 days post-MI. *n* = 3 to 4 in each group. (G) Immunohistochemistry and quantification for Emcn on control and *YAP/TAZ*-dKO heart sections. For numerical raw data, please see S1 Data. Scale bar 100μM. Acta2, smooth muscle actin alpha 2; Col1α1, collagen type I α 1; Col1α2, collagen type I α 2; Emcn, endomucin; Epi, epicardium; Lox, lysyl oxidase; LV, left ventricular; MI, myocardial infarction; Myo, myocardium; NS, non-significant; p4ha1, prolyl 4-hydroxylase subunit alpha-1; plod2, procollagen-lysine,2-oxoglutarate 5-dioxygenase 2; Plod2b, procollagen-lysine,2-oxoglutarate 5-dioxygenase 2b; qRT-PCR, real-time quantitative reverse transcription PCR; Tagln, transgelin; TAZ, transcriptional coactivator with PDZ-binding motif; Timp1, tissue inhibitor matrix metalloproteinase 1; Vcam1, vascular cell adhesion molecule 1; Vegf, vascular endothelial growth factor; YAP, yes-associated protein; YAP/TAZ-dKO, YAP/TAZ double knockout.

procollagen-lysine,2-oxoglutarate 5-dioxygenase 2 (*Plod2*), *Plod2b*, and lysyl oxidase (*Lox*)) and remodeling (tissue inhibitor matrix metalloproteinase 1 (*Timp1*) in *YAP/TAZ*-deficient hearts compared with controls at both days 7 and 28 post-MI (Fig 6E). Interestingly, the expression of *Col1α2*, *Axin2*, and *Timp1* was decreased in *YAP/TAZ*-dKO hearts only on day 7, but not at day 28 post-MI. Furthermore, to understand whether *YAP/TAZ* deficiency affects cardiac hypertrophy, we measured cardiomyocyte cross-sectional area and determined the expression of fetal genes. We found a reduced cardiomyocyte size in both border and remote zone of *YAP/TAZ*-dKO infarcted heart compared to controls as evident by wheat germ agglutinin (WGA) staining at 4 weeks post-MI (S18C and S18D Fig). Further supporting evidence showed a decrease in hypertrophy gene levels of myosin heavy chain 7 (*Myh7*), natriuretic peptide A (*Nppa*), and natriuretic peptide B (*Nppb*) in *YAP/TAZ*-dKO infarcted heart, suggesting an improvement in cardiac hypertrophy after MI (S18E Fig). These data suggest that both pro-inflammatory and reparative macrophage phenotypes after MI are YAP/TAZ dependent. Specifically, *YAP/TAZ* deficiency reduces macrophage-mediated pro-inflammatory response and adverse remodeling to preserve the cardiac function after MI.

To understand the mechanism of the vascular repair process after MI, we examined whether the expression level of angiogenic factors was affected in *YAP/TAZ*-deficient hearts. We observed significantly increased expression of *Vegf* and vascular cell adhesion molecule 1 (*Vcam1*) in *YAP/TAZ*-dKO hearts at day 7 post-MI (Fig 6F). We also observed a similar increasing trend at day 28 post-MI. However, the differences were not statistically significant. To further explore the process of neovascularization, we performed endomucin (Emcn) and CD31 immunohistochemistry on control and *YAP/TAZ*-dKO hearts at day 28 post-MI. Compared with control, we observed increased expression of Emcn and CD31, markers for endothelial cells of the capillary network, in the border zone of the infarcted myocardium of *YAP/TAZ*-dKO mice (Fig 6G and S19A Fig). Consistently, capillary density was also significantly increased in *YAP/TAZ*-dKO infarcted hearts compared with controls as evidenced by lectin staining (S19B Fig). Collectively, these results demonstrate that *YAP/TAZ* deficiency leads to improved vascularization of the infarcted myocardium. This may subsequently contribute to improved cardiac function recovery observed in *YAP/TAZ*-dKO mice.

## YAP activation promotes pro-inflammatory and inhibits reparative macrophage phenotype

To further establish the role of YAP in macrophage polarization, we activated YAP in myeloid cells using a conditional Rosa26 allele, *R26^YAP5SA,*^ and *LysM^Cre^* mice. This allele enables the in vivo expression of a constitutively active form of YAP (Fig 7A). YAP^5SA^ contains 5 canonical large tumor suppressor kinase (Lats) phosphorylation sites mutated from serine to alanine that

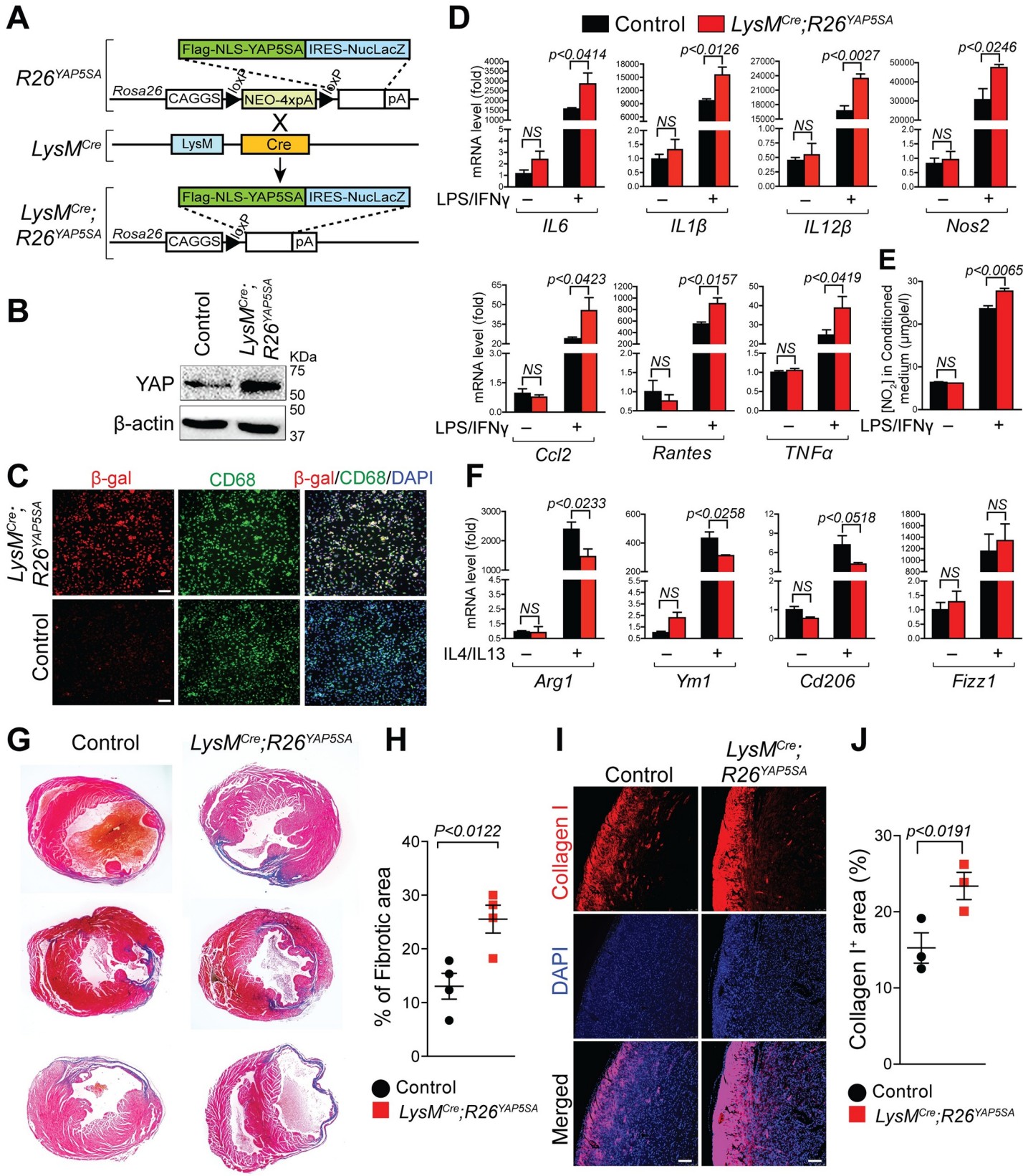

**Fig 7. YAP activation enhances pro-inflammatory but inhibits reparative macrophage phenotype.** (A) Generation of transgenic mice expressing YAP5SA, a constitutively active form of YAP in macrophages by crossing *Rosa26$^{YAP5SA}$* mice with *LysM$^{Cre}$* mice. (B) Western blot analysis for YAP was performed using total lysates from control and *LysM$^{Cre}$;Rosa26$^{YAP5SA}$* BMDMs. β-actin is shown as a loading control. (C) Double immunofluorescence for β-galactosidase (red) and CD68 (green) was performed on control and *LysM$^{Cre}$;Rosa26$^{YAP5SA}$* BMDMs. Nuclei were visualized by DAPI staining (blue). Scale bar 50 μM. (D–F) BMDMs were isolated from control and *LysM$^{Cre}$;Rosa26$^{YAP5SA}$* mice and stimulated with/without LPS/IFNγ or IL4/IL13 for 24 hours. (D) qRT-PCR for pro-inflammatory marker genes, *IL6*, *IL1β*, *IL12β*, *Nos2*, *TNFα*, *Ccl2*, *Rantes*, and *Ccr7* using RNA isolated from untreated or LPS/IFNγ-treated control and *LysM$^{Cre}$;Rosa26$^{YAP5SA}$* BMDMs. *n* = 3 in each group. (E) NO production determined by nitrite (NO$_2$−) levels in conditioned medium prepared from untreated or LPS/IFNγ-treated control and *LysM$^{Cre}$; Rosa26$^{YAP5SA}$* BMDMs. *n* = 3 in each group. (F) qRT-PCR for reparative marker genes, *Arg1*, *Ym1*, *Cd206*, and *Fizz1* using RNA isolated from untreated or IL4/IL13-treated control and *LysM$^{Cre}$;Rosa26$^{YAP5SA}$* BMDMs. *n* = 3 in each group. Gene expression results were normalized to *gapdh*, and results are represented as fold change. (G) Masson's trichrome staining on control and *LysM$^{Cre}$;Rosa26$^{YAP5SA}$* hearts at 28 days post-MI. Fibrotic area was normalized to the remaining heart for each heart section. (H) Quantification of Masson's trichrome stained fibrotic area in control and *LysM$^{Cre}$;Rosa26$^{YAP5SA}$* hearts at 28 days post-MI (*n* = 4 per group). (I) Immunostaining showing the expression collagen type I in infarct zone on control and *LysM$^{Cre}$;Rosa26$^{YAP5SA}$* heart sections at 28 days post-MI. DAPI was used to stain nuclei. Scale bar 100 μM. (J) Quantification of collagen type I positive area in control and *LysM$^{Cre}$;Rosa26$^{YAP5SA}$* heart sections. *n* = 3 per group. For numerical raw data, please see S1 Data. Arg1, Arginase-I; BMDMs, bone marrow–derived macrophages; Ccl2, C–C motif chemokine ligand 2; Cd206, cluster of differentiation 206; Fizz1, found in inflammatory zone 1; IFNγ, interferon gamma; IL, interleukin; IRES, internal ribosome entry site; LPS, lipopolysaccharide; NO, nitric oxide; Nos2, nitric oxide synthase 2; NS, non-significant; qRT-PCR, real-time quantitative reverse transcription PCR; TNFα, tumor necrosis factor alpha; YAP, yes-associated protein.

halt Hippo/Lats-mediated degradation of YAP [57]. *R26$^{YAP5SA}$* and *LysM$^{Cre}$;R26$^{YAP5SA}$* mice are referred to as control and *LysM$^{Cre}$;R26$^{YAP5SA}$*, respectively. YAP overexpression was confirmed by western blot analysis (Fig 7B). In addition to YAP$^{5SA}$, the *R26$^{YAP5SA}$* allele also carries a carboxyl-terminal internal ribosome entry site (IRES)-nuclear LacZ tag. The expression of the *R26$^{YAP5SA}$* transgene in BMDMs was further confirmed by β-gal and CD68 double immunostaining. Nuclear β-gal staining is observed in CD68 positive macrophages (Fig 7C). To determine the molecular changes associated with YAP activation, we isolated BMDMs from control and *LysM$^{Cre}$;R26$^{YAP5SA}$* mice and treated them with/without LPS/IFNγ or IL4/IL13. In unstimulated conditions, no significant changes in the expression of pro-inflammatory and reparative macrophage marker genes were detected between control and *LysM$^{Cre}$; R26$^{YAP5SA}$* BMDMs (Fig 7D–7F). However, upon LPS/IFNγ treatment, *LysM$^{Cre}$;R26$^{YAP5SA}$* BMDMs displayed a significant increase in pro-inflammatory gene expressions (*IL6*, *IL1β*, *IL12β*, *Nos2*, *TNFα*, *Ccl2*, *Rantes*, and *Ccr7*) *as* compared with control BMDMs (Fig 7D). Also, increased NO levels were detected in the condition media isolated from *LysM$^{Cre}$;R26$^{YAP5SA}$* BMDMs treated with LPS/IFNγ (Fig 7E). However, IL4/IL13 treated *LysM$^{Cre}$;R26$^{YAP5SA}$* BMDMs exhibited a decrease in the expression of reparative genes (*Arg1*, *Ym1*, and *Cd206*) when compared with control BMDMs (Fig 7F). Together, these data demonstrate that YAP activation can stimulate pro-inflammatory and inhibit reparative macrophage phenotype. Next, we investigated whether gene expression changes observed due to YAP activation in macrophages has any impact on cardiac fibrosis following MI. Masson's trichrome staining showed increased fibrotic area in *LysM$^{Cre}$;Rosa26$^{YAP5SA}$* hearts compared with controls at 28 days post-MI (Fig 7G and 7H). Similar to increased fibrosis, expression of collagen type I was also more pronounced in *LysM$^{Cre}$;Rosa26$^{YAP5SA}$* hearts compared to controls (Fig 7I and 7J) suggesting that cardiac remodeling is adversely affected leading to more fibrotic tissues and excess accumulation of extracellular matrix after MI.

## Discussion

Macrophages are present virtually in every tissue of the body, either as a resident or monocyte-derived cells. They infiltrate into tissues and provide immunity against pathogens, tissue development, homeostasis, and repair following damage. The robust inflammatory response after MI is essential for cardiac repair. However, timely suppression and containment of inflammation are also required for efficient and proper cardiac repair. Experimental studies have demonstrated that overactive or prolonged inflammatory response could lead to cardiomyocytes death, impaired systolic function, fibrosis, left ventricular dilatation, and heart failure

following MI. It is well established that macrophages are the predominant immune cells infiltrating the infarcted myocardium and modulating both pro-inflammatory and anti-inflammatory/reparative responses during the inflammatory and reparative phases of cardiac repair. Therefore, understanding the dynamics of macrophage polarization will ultimately help to modulate post-MI inflammation, resolution, and remodeling, and improve the heart's ability to repair itself after injury. Here, we demonstrate that the Hippo signaling mediators YAP and TAZ can act both as transcriptional coactivators or corepressors during pro-inflammatory and reparative macrophage polarization, respectively, and modulate post-MI inflammation and remodeling by promoting pro-inflammatory and limiting reparative macrophage phenotype.

Several preclinical studies have demonstrated the involvement of macrophages in MI pathophysiology [1–3]. The initial pro-inflammatory phase after MI is dominated by the recruitment of pro-inflammatory macrophages that exhibit phagocytic, proteolytic, and inflammatory functions. Here, we demonstrate that *YAP/TAZ* are essential for the macrophage-mediated initial inflammatory response. In addition, our RNA-seq analysis uncovers numerous YAP/TAZ-dependent transcriptional changes that may play an important role in different inflammatory diseases. Our global and candidate-based approaches demonstrate that pro-inflammatory genes are positively regulated by YAP/TAZ. For example, in addition to other pro-inflammatory genes, both *IL6* and *IL1β* expressions are reduced in *YAP/TAZ*-deficient BMDMs stimulated with LPS/IFNγ, as well as in *YAP/TAZ*-deficient macrophages from the hearts after MI. Consistently, *IL6* and *IL1β* expressions are elevated in BMDMs of transgenic mice expressing YAP5SA, a constitutively active form of YAP in macrophages. We further demonstrate that YAP/TAZ can promote IL6 expression not only by directly binding to TBSs in the *IL6* promoter but also through the Tak1-p38 MAPKs signaling pathway. However, the YAP/TAZ-mediated direct regulation of *IL6* promoter seems to play a more dominant role. YAP activation has been associated with increased IL6 production in previous studies [58,59]. However, a direct regulation was not established. Consistent with our findings, recently, Zhou and colleagues also identified TEAD-binding sites in human IL6 promoter and showed direct regulation using a human leukemia monocytic cell line (THP-1 cells) [45]. We also observed decreased number of pro-inflammatory (F4/80+iNOS+) macrophages and increased number of reparative (F4/80+CD206+) macrophages in the *YAP/TAZ*-deficient heart after MI suggesting that YAP/TAZ promote pro-inflammatory and limit reparative macrophage fate. Several preclinical and clinical studies have shown that targeting pro-inflammatory macrophages to suppress their pro-inflammatory response post-MI is cardioprotective. For example, interferon regulatory factor 5 (IRF5) knockdown in macrophages shifts pro-inflammatory to reparative phenotype to facilitate inflammation and improve cardiac healing post-MI [25,28]. Similarly, knockdown of *Ccr2* decreases macrophages recruitment to the infarct site and reduces the pro-inflammatory response and infarct size [27]. Modulation of recruited or resident cardiac macrophages from pro-inflammatory to anti-inflammatory state using phosphatidylserine-presenting liposomes at a predetermined time after MI also reduces adverse remodeling of the heart [24]. Consistent to these findings, our study also demonstrates that genetic deletion or pharmacological inhibition of YAP/TAZ in macrophages reduces the pro-inflammatory response, cardiac fibrosis, and protects cardiac function after MI suggesting that YAP/TAZ can be considered as therapeutic targets to modulate macrophage phenotype in diseased conditions.

In contrast to the early pro-inflammatory phase, the reparative/anti-inflammatory phase is dominated by reparative macrophages as they actively participate in cardiac repair [24,60–63]. Previous studies have demonstrated that reparative macrophages facilitate repair response by promoting myofibroblasts formation, neo-angiogenesis, and anti-inflammatory functions [3,14–16,64]. The depletion of reparative macrophages leads to heart failure due to the lack of

resolution of inflammation. This is evident by the increased number of inflammatory cells in the infarct tissue and impaired left ventricular contractile function [7,65]. Here, we demonstrate that YAP/TAZ inhibit reparative phenotype by repressing expression of reparative genes such as *Arg1*. Expression of reparative genes such as *Arg1*, *Ym1*, *Egr2*, and *Cd206* are increased in *YAP/TAZ*-deficient BMDMs stimulated with IL4/IL13, as well as in *YAP/TAZ*-deficient macrophages from the hearts at day 7 post-MI. Consistently, expression of *Arg1* and other reparative genes are decreased in BMDMs of transgenic mice expressing YAP$^{5SA}$, a constitutively active form of YAP in macrophages. In contrast to the recent finding by Zhou and colleagues that YAP expression is reduced during reparative macrophage polarization, we observed increased YAP, pYAP, and TAZ expression after IL4/IL13 treatment in RAW264.7 cells, PMs, and BMDMs [45]. We further demonstrate that YAP/TAZ represses *Arg1* expression by directly binding to the promoter and recruiting HDAC3-NCoR1 repressor complex. We also demonstrate that YAP/HDAC3-mediated repression of *Arg1* expression is independent of the deacetylase activity of HDAC3. Both YAP and TAZ are well-established transcriptional coactivators. However, their functions as transcriptional corepressors are not well described. Two recent reports have demonstrated that YAP/TAZ recruits the nucleosome remodeling and deacetylase (NuRD) complex to repress gene expression [66,67]. However, YAP/TAZ interaction with the HDAC3-NCoR1 repressor complex has not been reported. HDACs are recruited to the target gene promoters via their interaction with transcription factors, cofactors, and multiprotein coregulatory complexes often referred to as coactivators or corepressors. For example, HDAC3 interacts with 2 major corepressor complexes, nuclear receptor corepressor (NCoR) or its homolog silencing mediator of retinoic and thyroid receptors (SMRT) to drive constitutive repression. A recent study suggests that HDAC3's interaction with NCoR/SMRT is essential for transcriptional repression and not its deacetylase activity [55]. Similar to YAP/TAZ, HDAC3 is also required for promoting inflammatory gene expression as *HDAC3* deletion leads to reduced expression of LPS-induced pro-inflammatory genes including *IL6* [68]. Moreover, HDAC3 negatively regulates reparative macrophage polarization, evident by increased expression of IL4-induced reparative genes including *Arg1* in the BMDMs derived from macrophage-specific *HDAC3* knockout mice [69]. In contrast to some of the reparative gene analyzed, *HDAC3* deletion did not alter the H3K9Ac or H3K27Ac marks around the HDAC3 binding regions in *Arg1* promoter suggesting that the deacetylase activity of HDAC3 is not essential for *Arg1* repression [69]. Similarly, macrophage-specific deletion of *NCoR* causes impaired inflammatory response and increased anti-inflammatory response resembling the *HDAC3* knockout mice, demonstrating that HDAC3 interaction with NCoR is essential for its role in macrophage polarization [70]. A recent study by Zhou and colleagues demonstrated that YAP inhibits reparative macrophage polarization by promoting expression of transcriptional factor p53, a negative regulator of reparative macrophage polarization [45]. p53 expression is increased during both pro-inflammatory or reparative macrophage polarization but more prominent in pro-inflammatory macrophages compared to reparative macrophages [49]. Previous studies have demonstrated that p53 negatively regulates activation of both pro-inflammatory (*IL6*, *TNFα*, etc.) and reparative (*Arg1*, *Fizz1*, *Irf4*, etc.) marker genes [49,71,72]. In contrast to the finding by Zhou and colleagues, we did not observe any significant changes in p53 expression in *YAP/TAZ*-deficient BMDMs during pro-inflammatory or reparative macrophage polarization. Together, our results suggest that depending upon the biological context YAP/TAZ can act as transcriptional coactivators or corepressors [67].

Reparative macrophages also secrete anti-inflammatory and pro-fibrotic cytokines (e.g., IL-10 and TGF-β) to suppress inflammation and promote tissue repair. These cytokines by modulating (myo)fibroblast activation and collagen deposition [1]. IL-10 is produced by

macrophages as a negative feedback mechanism to dampen the production of inflammatory cytokines and minimize the deleterious effects after MI. We observed decreased expression of both *IL-10* and *TGF-β* in *YAP/TAZ*-deficient hearts at day 7 post-MI. Our results are consistent with a previous study demonstrating improved cardiac dysfunction and cardiac fibrosis in macrophage-specific *IL-10* knockout mice [1]. Reduced fibrosis in post-MI *YAP/TAZ*-dKO hearts is also associated with reduced formation of myofibroblasts. This is evident by decreased expression of *Acta2* and *Tagln* marking stress fibers. Additionally, genes essential for collagen synthesis and its processing were also down-regulated in *YAP/TAZ*-dKO hearts. We speculate that reduced myofibroblasts activation and collagen production result from reduced expression of *IL-10* and *TGF-β* in *YAP/TAZ*-dKO hearts and early shift of pro-inflammatory to reparative phenotype. Both processes lead to reduced tissue damage and adverse remodeling. Previous findings suggest that macrophage-mediated reparative/anti-inflammatory signaling alone determines the infarct repair [63]. However, our results demonstrate that both YAP/TAZ-dependent pro-inflammatory and reparative/anti-inflammatory responses are essential for limiting the infarct size and repair.

In addition to regulating pro-inflammatory and reparative responses, macrophages enhance neovascularization by secreting a variety of pro-angiogenic growth factors including *Vegf* and *Vcam1*. Using a reporter mouse line, Willenborg and colleagues demonstrated that *Vegf* expression by macrophages is required to induce capillary sprouting during wound healing [73]. Recent studies have shown that signals derived from macrophages are necessary to drive angiogenesis during cardiac regeneration [4,5]. Consistent with these findings, we also observed enhanced expression of *Vegf* and *Vcam1* in *YAP/TAZ*-dKO post-MI hearts, suggesting improved revascularization of the infarct tissue. Increased number of CD31 and Emcn-positive blood vessels as well as increased lectin-positive capillary density in the infarcts of *YAP/TAZ*-dKO hearts support this notion. Improved cardiac hypertrophy is associated with cardiac recovery after MI. Here, we observed a reduced expression of hypertrophy genes (such as *Myh7*, *Nppa*, and *Nppb*) and cardiomyocyte size in *YAP*/*TAZ*-dKO infarcted heart compared to controls as evident by WGA, indicating an improvement of cardiac remodeling post-MI. Collectively, these findings demonstrate that the deletion of *YAP/TAZ* in macrophages reduces inflammation and enhances repair, while simultaneously promoting angiogenesis in the injured cardiac tissue.

Growing evidence suggests that an early shift in macrophage polarization from the pro-inflammatory phase to the reparative phase is essential for successful cardiac repair [24,25,27,28]. In the present study, we demonstrate that YAP/TAZ are essential for regulating macrophage polarization and post-MI response. Deletion of *YAP/TAZ* attenuates early inflammatory response and promotes timely polarization of pro-inflammatory macrophages to reparative macrophages. These changes lead to mitigated adverse remodeling and improved cardiac function after MI. Our results demonstrate that YAP/TAZ are promising targets for modulating macrophage polarization and improving cardiac repair after MI. Since macrophages can take up foreign particles, they are easy and selective targets. This represents a precious opportunity for therapeutic intervention to prevent adverse cardiac remodeling and physiological deterioration of the infarcted hearts.

## Materials and methods

### Ethics statement

All animal procedures were approved (IACUC protocol No. 2019/SHS/1461 and 2018/SHS/1415) by the Institutional Animal Care and Use Committee at Duke-NUS Medical School/Singhealth conforming to the Guide for the Care and Use of Laboratory Animals (National Academies Press, 2011).

## Mice

Myeloid cell-specific *YAP/TAZ*-dKO mice were generated by crossing the *LysM^Cre/+* transgenic mice with *YAP^flox/flox;TAZ^flox/flox* mice [74–76]. The resulting *LysM^Cre/+;YAP^flox/+;TAZ^flox/+* offspring were then backcrossed to *YAP^flox/flox;TAZ^flox/flox* strain to obtain *LysM^Cre/+;YAP^flox/flox; TAZ^flox/flox* (*YAP/TAZ*-dKO) mice. We used *YAP^flox/flox;TAZ^flox/flox* mice as control. *YAP* and *TAZ* floxed mice were genotyped as described previously [74,75]. Similarly, for YAP activation in myeloid cells, *LysM^Cre/+* mice were crossed with *Rosa26^YAP5SA/YAP5SA* knock-in strain to generate *LysM^Cre/+;Rosa26^YAP5SA/+* mice [77]. We used *Rosa26^YAP5SA/+* mice as control. *Rosa26^YAP5SA/YAP5SA* mice were genotyped as described previously [77]. All mice were maintained on a mixed (C57BL/6 and Sv/129) genetic background. Both male and female mice were used in the experiments. All animals were kept in ventilated cages (up to 5 mice per cage) in a 12-hour light–dark cycle and were provided water and food at all times. All animal procedures were approved (IACUC protocol No. 2019/SHS/1461 and 2018/SHS/1415) by the Institutional Animal Care and Use Committee at Duke-NUS Medical School/Singhealth conforming to the Guide for the Care and Use of Laboratory Animals (National Academies Press, 2011).

## Myocardial infarction surgery and echocardiography

MI surgery was done as described previously [78]. Briefly, control and *YAP/TAZ*-dKO mice were subjected to permanent ligation of left anterior descending (LAD) coronary artery to induce MI in aseptic conditions. At first, mice were anesthetized with isoflurane, fixed in a supine position with tape. Their neck and chest were shaved and applied with an antiseptic solution. Toe pinch reflexes were performed to observe the depth of anesthesia. The tongue was retracted, and a 20-gauge IV catheter was quickly inserted into the trachea. The catheter was connected to the small animal ventilator through a Y-shaped connector. Mice were then ventilated with an adjusted stroke volume and a respiratory rate of 100 to 110 breaths/min. Next, thoracotomy was performed, and the suture was inserted through the LV of exposed heart. To induce MI, a permanent ligation was obtained by tying an 8–0 silk suture around the LAD. The echocardiographic assessment was conducted at 28 days post-MI to evaluate the cardiac function. Hearts were harvested at 2, 3, 6, 7, and 28 days post-MI for histology, qRT-PCR, FACS, and immunohistochemistry. Transthoracic echocardiography (TTE) was conducted with Vevo 2100 (VisualSonics, VSI, Toronto, Canada) by using a MS400 linear array transducer, 18 to 38 MHz. All mice were anesthetized with 5% isoflurane and maintained at 0.6% to 1%. An average of 10 cardiac cycles of standard 2-dimensional (2D) parasternal long and short axis (mid papillary muscle level), as well as pulsed-wave Doppler recording of the mitral inflow, were acquired and stored for subsequent offline analysis. LVEDV and LVESV were determined at the parasternal long axis. SV was obtained by subtracting ESV from EDV. Left ventricular internal diameter at end-systole (LVIDes) and end-diastole (LVIDed) were recorded from the 2D guided M-mode of the short axis at the mid papillary muscle level. Subsequently, LVEF and FS were calculated using the modified Quinone method as described before [79,80]. IVRT was obtained with pulsed-wave Doppler of the mitral valve at the apical 4 chamber view and was corrected for heart rate (IVRTc) as calculated to the formula (IVRT = IVRT$/\sqrt{RR\%}$). All measurements were averaged on 3 cardiac cycles by a blinded operator.

## Histology and immunohistochemistry

Histology and immunohistochemistry were performed as described previously. Briefly, adult hearts were dissected in PBS and fixed in 4% paraformaldehyde overnight at 4°C. Hearts were then washed with PBS, dehydrated in an ethanol series, and stored in 100% ethanol at −-20°C.

Paraffin sections were prepared for Masson's trichrome, WGA, MPO, CD31, Lectin, and Emcn immunostaining. Collagen immunostaining was performed on cryosections. For immunostaining, sections were permeabilized with 0.1% Triton-X-100 in PBS and processed for antigen retrieval using antigen unmasking solution (Vector Laboratories, United States of America). Endogenous peroxidase activity was blocked with 3% H2O2 10 minutes at room temperature. The cryo/paraffin heart sections were blocked for 1.5 hours using 5% BSA or 10% normal goat/donkey serum. After washing with PBS, sections were incubated with primary antibodies diluted in blocking buffer at 4°C overnight. Sections were washed and then incubated with secondary antibody (HRP-conjugated or Alexa Fluor conjugated) diluted in blocking buffer for 1.5 hours at room temperature. Sections were washed again and developed using the DAB kit (Vector Laboratories, catalog no. SK-4100) or DAPI (for IF) in PBS for 15 minutes. After washing, sections were mounted with 100% glycerol, and the staining pattern was visualized with a Leica fluorescence microscope (LEICA DMi8, Leica Microsystems (SEA), Singapore).

## Cell culture

BMDMs were isolated from the femurs and tibiae of control and *YAP/TAZ*-dKO mice. Femurs and tibiae were collected by cutting off the hind legs at the hip joint with sterile scissors, ensuring that joints were kept intact, and placed in sterile cold PBS. The excess muscle was removed carefully by leaving the bone with no muscle. Using sharp scissors soaked in 95% ethanol, the ends of the bones were cut. The bone cavity containing BMDMs was flushed out with 5 ml of sterile cold PBS. Cells were then centrifuged at 1,000 rpm for 10 minutes for pellet collection. Pellets were resuspended and plated in 20 ml of macrophage medium [Dulbecco's Modified Eagle Medium (DMEM) (Gibco, Thermo Fisher Singapore, catalog no. 11965092) containing 20% L929-conditioned medium, 1% penicillin/streptomycin, and 10% FBS]. On day 3, an additional 15 ml of macrophage medium was added, and on day 6, all the media were replaced with fresh macrophage media. On day 7, cells were detached using trypsin (Gibco, catalog no. 25200056) for 30 minutes to 1 hour at 4°C. Cells were then washed, centrifuged, resuspended and seeded ($1 \times 10^6$ cells/well) in a 6-well tissue culture plates using the culture medium (DMEM containing 10% FBS and 1% penicillin/streptomycin). CD11b-FITC (Miltenyi Biotec, Germany, catalog no. 130-098-085) and APC anti-mouse F4/80 (BioLegend, USA, catalog no. 123115) antibody-based flow cytometry were used to evaluate the macrophage (BMDMs and PMs) culture purity. We observed nearly 99.69% and 97.63% purity for BMDMs and PMs, respectively. Also, CD68 immunostaining was performed to evaluate the macrophage culture purity. BMDMs were subsequently treated with pro-inflammatory (LPS 100 ng/ml + IFNγ 10 ng/ml), reparative/M2a stimuli (IL4 10 ng/ml + IL13 10 ng/ml), reparative/M2b (100 μg/ml anti-BSA + 10 μg/ml BSA), and reparative/M2c (IL10 10 ng/ml + TGFβ1 10 ng/ml) for desired time points with appropriate controls. IFNγ (catalog no. 575306), IL4 (catalog no. 574304), IL13 (catalog no. 575902), and TGFβ1 (catalog no. 763102) were obtained from BioLegend and LPS (catalog no. L3129) from Sigma-Aldrich, Singapore. IL10 (catalog no. ab9736) was obtained from Abcam, USA, and anti-BSA (catalog no. SAB4301142) was obtained from Merck, Singapore. PMs were harvested from wild-type mice treated with an intraperitoneal injection of 3% Brewer thioglycollate medium. PMs were harvested 4 days after thioglycollate injection and cultured and maintained using conditions similar to BMDMs, described above.

## RNA interference

Control (AllStars Negative siRNA AF488, catalog no. 1027284), *YAP* siRNAs (catalog no. GS22601), and *TAZ* siRNA (catalog no. SI01442511 and SI01442532) were obtained from

Qiagen, USA. HDAC3 silencer pre-designed siRNA (catalog no. AM16708) and silencer nega-tive control siRNA (catalog no. AM4620) were purchased from Invitrogen, USA. The siRNA was transfected in BMDMs using Lipofectamine RNAiMAX transfection reagent (Thermo Fisher, Singapore, catalog no. 13778150). Briefly, after a transient transfection, BMDMs were incubated for 72 hours and treated with either pro-inflammatory (LPS 100 ng/ml + IFN$\gamma$ 10 ng/ml) or reparative stimuli (IL4 10 ng/ml + IL13 10 ng/ml) diluted in culture medium (DMEM containing 10% FBS and 1% penicillin/streptomycin). Cells were collected for total RNA isolation and qRT-PCR analysis using Favorgen RNA extraction kit (Favorgen Biotech, Taiwan, catalog no. FATRK 001–2). Cell lysates were prepared using RIPA buffer (Thermo Fisher, catalog no. 89901) for western blot analysis.

## Macrophage migration assay

BMDMs ($1 \times 10^5$ cells) from control and *YAP/TAZ*-dKO mice were added to the upper cham-ber of a 6.5-mm Transwell plate system with 8-μm pore polycarbonate membrane insert. The lower chamber of the Transwell plate was loaded with macrophage medium containing LPS/IFN$\gamma$, and the BMDMs were allowed to migrate for 12 hours. The cells that migrated into the lower chamber media were fixed with 4% PFA and subsequently stained with CD68 and DAPI and quantified with ImageJ software (NIH, USA).

## Neutrophil isolation and migration assay

Control and *YAP/TAZ*-dKO mice were injected intraperitoneally with a dose of LPS (2 mg/kg) as diluted in PBS. Neutrophils were isolated from mouse bone marrow at 16 hours post-injec-tion as described previously [81]. Neutrophils ($1 \times 10^4$ cells) from control and *YAP/TAZ*-dKO mice were added to the upper chamber of a 6.5-mm Transwell plate system with 8-μm pore polycarbonate membrane insert. The lower chamber of the Transwell plate was loaded with neutrophil medium containing RPMI 1640 supplemented with 10% FBS and 1% penicillin/streptomycin. Neutrophils were allowed to migrate for 6 hours, and the cells that migrated into the lower chamber were fixed with 4% PFA and subsequently stained with myeloperoxi-dase (MPO) and DAPI and quantified with ImageJ software.

## Plasmids

Mouse *IL6* promoter (approximately 2 kb) was amplified and cloned into a pGL4.27 vector (Promega, Singapore) for luciferase assays. For mutation of TBSs in *IL6* promoter, 3 promoter fragments (Pro-I, Pro-II, and Pro-III) were amplified and cloned into pGL4.27 vector and used as control. Mini genes for the same promoter fragments differing only by the presence of mutated TBSs were purchased from Integrated DNA Technologies (IDT, USA)) and subse-quently cloned into pGL4.27 vector. In Pro-I promoter fragment, TBSs AGAATG, TGCATG, and TGAATG were mutated to ACAAAC, TCCAAC, and TCAAAC, respectively. In Pro-II promoter fragment, TBSs TAGTATG, AAGAATG, TGAATG, and TGTATG were mutated to TACTAAC, AACAAAC, TCAAAC, and TCTAAC, respectively. Similarly, in Pro-III pro-moter fragment, TBSs TGTATG, CGTATG, and AGCATG were mutated to TCTAAC, CCTAAC, and ACCAAC, respectively. Mouse YAP and TAZ expression vectors for the lucif-erase assay have been previously described [82]. Other plasmids used in this study were the fol-lowing: pCMV5-Flag-HDAC3 was a gift from Qingbo Xu and Lingfang Zeng (Addgene plasmid # 63676) [83]. pGL3-mArg1 promoter/enhancer −31/−3810 was a gift from Peter Murray (Addgene plasmid # 34571) [84]. pCMV-flag YAP$^{S127A}$ was a gift from Kunliang Guan (Addgene plasmid # 27370) [57]. NF$\kappa$B luciferase reporter (Promega, catalog no. E849A). HDAC3 and HDAC3$^{H134A,\ H135A}$ was a gift from Chinmay Trivedi [56].

## Luciferase assay

Luciferase assay was performed as previously described [41]. Briefly, HEK293T cells were maintained in DMEM supplemented with 10% FBS and 1% P/S. HEK293T cells were seeded in 12-well plates for 24 hours before transfection. The *IL6* or *Arg1* luciferase reporter plasmid and other indicated plasmids (YAP, TAZ, or HDAC3) were co-transfected in desired concentrations using FuGENE6 reagent (Promega, catalog no. E2691). To normalize transfection efficiency, 50 ng of lacZ expression plasmid was also transfected along with other indicated plasmids. Sixty hours post-transfection, cells were lysed and extracted with lysis buffer (Promega, catalog No. E3971). Next, luciferase activities were measured using the Luciferase Reporter Assay System kit (Promega, catalog no. E1500). Cell lysates were also evaluated for β-galactosidase activity using the β-Galactosidase Enzyme Assay System (Promega, catalog no. E2000). Luciferase reporter activity was then normalized to β-galactosidase activity. The luciferase assay was repeated in at least 3 independent experiments and duplicates. Representative data are shown in the figures. The inhibitors used in the luciferase assay were SB203580 (Selleck Chemicals, USA, catalog no. S1076), RGFP966 (Selleck Chemicals, catalog no. S7229), Vorinostat (Selleck Chemicals, catalog no. S1047), and Scriptaid (Sigma, catalog no. S7817). Luciferase assay with HDAC3 deacytylase mutant, lentiviral particles were prepared using HDAC3 and HDAC3$^{H134A, H135A}$ plasmids. HEK293T cells were seeded in 12-well plates for 24 hours before transfection. The *Arg1* luciferase reporter plasmid was co-transfected with YAP expression plasmid and 50 ng of lacZ expression plasmid for normalization. Twenty-four hours after transfection, cells were infected with either HDAC3 or HDAC3$^{H134A, H135A}$ lentivirus. Infected cells were harvested for luciferase reporter analysis 48 hours after infection using the Luciferase Reporter Assay System kit and β-Galactosidase Enzyme Assay System.

## Bacteria killing assay

To determine the bactericidal ability of macrophages, $2 \times 10^5$ gram-negative (*E. coli*) or $2 \times 10^5$ gram-positive (*L. monocytogenes*) bacteria were incubated with BMDMs for 2 hours. Cells were then thoroughly washed with PBS for 3 to 5 times and incubated for 16 to 24 hours in DMEM containing antibiotics before harvesting for intracellular bacteria. Subsequently, cell lysate from macrophages containing intracellular bacteria was serially diluted with PBS and spread onto agar plates to determine bacterial viability (expressed as CFU). For qRT-PCR analysis, total RNA was prepared using Favorgen RNA extraction kit (Favorgen Biotech, catalog no. FATRK 001–2).

## Flow cytometry and FACS

Infarcted LV tissues excised from day 2 (to isolate pro-inflammatory macrophages) and day 6 (to isolate reparative macrophages) post-MI mice were digested with 600 U/ml collagenase II (Thermo Fischer, catalog no. 17101015), DNase I 60 U/ml (Thermo Fischer, catalog no. EN0521), and 5% FBS in Hanks buffered saline solution (Thermo Fischer, catalog no. 14175095) at 37°C with shaking for 20 minutes. Cells suspension was then filtered through a 30-μM separation filter, and suspended cells were transferred to FACS tube. Isolated cells were centrifuged and washed with 2% FBS in PBS buffer and blocked with FcR blocking reagent (Miltenyi, catalog no. 130-092-575) for 15 minutes at 4°C. Next, cells were stained with PE-conjugated F4/80 (1:100; Invitrogen, catalog no. 12-4801-82), Alexa Fluor 448-conjugated iNOS (1:50; Santa Cruz, USA catalog no. sc-7271), and Alexa Fluor 700-conjugated CD206 (1:100; BioLegend, catalog no. 141734) antibodies to obtain pro-inflammatory and reparative macrophages. iNOS$^+$/F4/80$^+$ macrophages were considered as pro-inflammatory macrophages, and CD206$^+$/F4/80$^+$ macrophages were considered as reparative macrophages. Flow

cytometric analyses and macrophage sorting were performed by using a FACSAria II cell sorter (BD Bioscience, Singapore).

## In vitro nitrite level

Nitrite resulting from NO was examined in BMDMs conditioned medium with Griess Reagent System (Promega, catalog no. G2930) according to manufacturer's instructions.

## Protein extraction and western blot

Cells were washed twice with cold DPBS (Lonza, Switzerland, catalog no. 17-512F) and lysed with RIPA buffer (Thermo Scientific, catalog no. 89901) containing 1:100 diluted protease and phosphatase inhibitor cocktail (Sigma). The cell lysates were centrifuged at 13,000 rpm for 10 minutes at 4°C, and the supernatants were collected. Total protein concentration was determined with the Pierce BCA protein assay kit (Thermo Scientific, catalog no. 23225) following the manufacturer's instructions. Next, 20 to 30 μg of total protein samples were separated by SDS-PAGE and transferred to nitrocellulose membrane using Trans-Blot Turbo system (Bio-Rad, Singapore). Membranes were then blocked with 2% to 5% BSA in TBS containing 0.1% Tween (TBST) and subsequently incubated with primary antibodies diluted in TBST containing 2% to 5% BSA for overnight at 4°C. Blots were then washed in TBST and incubated for 1.5 hour at room temperature probed with the appropriate horseradish peroxidase linked IgG (Santa Cruz). Immunoreactive bands were detected by chemiluminescence (Hiss GmbH, Germany, catalogue no. 16026) using Gel Doc XR+ System (Bio-Rad).

## Antibodies

The following antibodies were used for western blotting and immunostaining: YAP (Santa Cruz, catalog no. sc-376830 and Sigma, catalog no. Y4770), pYAP (Ser127) (Cell Signaling, Catalog no. 4911), TAZ (Santa Cruz, catalog no. sc-48805 and Santa Cruz, Catalog no. sc-518026), IL6 (Santa Cruz, catalog no. sc-57315), Nos2 (Santa Cruz, catalog no. sc-7271), HDAC3 (Santa Cruz, catalog no. sc-376957), NCoR1 (Thermo Fisher, catalog no. PA1-844A), Myc (Sigma, Catalog no. C3956), FLAG (Sigma, Catalog no. F1804), IL1β (Santa Cruz, catalog no. sc-7884), Arg1 (Santa Cruz, catalog no. sc-166920), Collagen I (Novus Biologicals, USA, catalog no. NB600-408), Emcn (Santa Cruz, catalog no. sc-65495), CD31 (BD Bioscience, catalog no. 553370), Ki67 (Abcam, catalog no. ab16667), CD68 (AbD Serotec, United Kingdom, catalog no. MCA1957), TGFβ (Cell Signaling, Catalog no. 3711), β-galactosidase (MP Biomedicals, USA, catalog no. 863365), β-actin (Santa Cruz, catalog no. sc-47778), PCNA (Santa Cruz, catalog no. sc-56), WGA (Thermo Fisher, catalog no. W11261), Lectin (Thermo Fisher, catalog no. L21409), and Myeloperoxidase (Abcam, catalog no. ab9535).

## RNA isolation, cDNA synthesis, and qRT-PCR

Total RNA from heart tissues (left ventricles) and BMDMs were isolated using TRIzol (Life Technologies, Singapore, catalog no. 15596018) and Favorgen RNA extraction kit (Favorgen Biotech, catalog no. FATRK 001–2), respectively. The concentration and quality of RNA were measured with UV spectrophotometry (NanoDrop Technologies, Wilmington, North Carolina, USA). For cDNA synthesis, total RNA was reverse transcribed using random hexamers and SuperScript III FirstStrand Synthesis system (Life Technologies, catalog no. 18080–051). Gene expression was performed by quantitative RT-PCR (ABI PRISM 7900 or ViiA7 Real-Time PCR System) using 10-μl reaction mixture containing 20 ng cDNA, SYBR Green Master Mix (Life Technologies, catalog no. 4368702), 6-μM forward primer, and 6-μM reverse primer.

Results were analyzed with Real-Time PCR System Software (Applied Biosystems, Singapore). All mRNA data were normalized against the reference gene Gapdh.

## RNA-seq

To perform the RNA-seq, total RNA was prepared from BMDMs of control and *YAP/TAZ-dKO* mice using either Favorgen RNA extraction kit (Favorgen Biotech, catalog no. FATRK 001–2) or PureLink RNA Mini Kit (Thermo Fisher, catalog no. 12183018A). Subsequently, strand-specific poly(A) RNAseq libraries were generated using NEBNext Ultra II Directional RNA Library Prep Kit for Illumina (NEB,Singapore, catalog no. E7765S) and sequenced on Hiseq2000 sequencer. The average sequencing depth was >30 million reads per sample. Paired-end RNA sequencing reads from each sample were aligned to the mouse reference genome (GRCm38) via STAR aligner (version 2.5.1b) with an average mapping rate of 81% [85]. Mapped sequencing reads were quantified and assigned to genomic features via the featureCount option in R package Rsubread [86]. Differential gene expression analysis was conducted in R via limma [87]. Genes with absolute log fold change >1.0 and false discovery rate (FDR) <5% were considered differentially expressed. Common and unique sets of differentially expressed genes were identified through Venn diagrams (http://bioinfogp.cnb.csic.es/tsools/venny/index.html). Transcriptome data were further analyzed for enrichment of biological pathways by querying "gene-sets" from the Kyoto Encyclopedia of Genes and Genomes (KEGG) [88] or Gene Ontology [89] pathway repositories via the PreRankedGSEA [90] or Enrichr overrepresentation analysis tools [91].

## ChIP assay

ChIP assay was performed as previously described [41]. In brief, BMDMs were stimulated with LPS/IFNγ (100 ng/ml + 10 ng/ml) or IL4/IL13 (10 ng/ml) for 12 hours and subsequently cross-linked for 10 minutes with 1% formaldehyde at room temperature. Cells were then collected in ice-cold PBS and processed for ChIP using Millipore Assay Kit (Millipore, Germany, catalog no. 17–295). ChIP grade YAP and TAZ antibody were used for the assay (Cell Signaling, catalog no. 14074S, and 4883S). Rabbit IgG was used as a negative control. *IL6* promoter PCR was performed with the following primers (sense: GGTGAAAGAATGGTGGACTCA; antisense: CATTCTCCCCAGTGGTCTCT). *Arg1* promoter PCR was performed with the following primers (sense: GGAGGCAGGCGATACTTTAAT; antisense: ACAGACTCCCCACGTTACCA).

## Co-immunoprecipitation

To examine the protein–protein interaction, RAW 264.7 cells were transfected with plasmid expressing Myc-YAP and FLAG-HDAC3 using lipofectamine transfection reagent (Thermo Fischer, catalog no. 11668019) according to the manufacturer's protocol. Following transient transfection, cells were incubated for 48 hours and starved for 8 hours with serum-free DMEM supplemented with 1% penicillin/streptomycin. Cells were then treated with reparative macrophage-stimuli (IL4 10 ng/ml + IL13 10 ng/ml) for 12 hours. Cell lysates were collected for co-immunoprecipitation using protein extraction reagent. The cells extracts were then incubated with anti-Myc (1:100; Sigma, Catalog no. F1804) at 4°C overnight, and immunoprecipitation was performed using protein A/G-agarose beads immunoprecipitation reagents (Santa Cruz, catalog no. sc-2003). Samples were then run on SDS-PAGE and probed with anti-FLAG (1:2000; Sigma, Catalog no. F1804) and anti-Myc (1:2000; Sigma, Catalog no. C3956), antibodies at 4°C overnight to detect them as co-precipitated protein. In a separate experiment, BMDM cells were treated with reparative macrophage-stimuli (IL4 10 ng/ml + IL13 10 ng/ml)

for 12 hours and performed co-immunoprecipitations with anti-YAP (1:50, Cell Signaling, Catalog no. 14074S) and immunoblotted for NCoR1 (1: 500; Thermo Fisher, catalog no. PA1-844A), anti-HDAC3 (1: 500; Santa Cruz, catalog no. sc-376957), and anti-YAP (1: 1000; Sigma, catalog no. Y4770). The immune complexes were detected using ECL reagent (GE Healthcare, USA, catalog no. RPN2235).

## Fibrosis image analysis

The amount of post-MI myocardial fibrosis in control and mutant mice was quantified using ImageJ. Briefly, serial sections starting from the coronary ligature to the cardiac apex were stained with Masson's trichrome to determine the fibrotic tissue. At least 5 cross-sectional images for each heart were analyzed for blue myocardium (fibrotic tissue), followed by the total heart. Statistical differences in the percentage of fibrosis were determined with the unpaired $t$ test.

## Statistical analysis

Experimental data are presented as mean ± standard error mean (SEM) from at least 3 independent experiments. Data from luciferase experiments are presented as mean ± standard deviation (SD). Results were analyzed with unpaired $t$ test within 2 groups if both groups were normally distributed. Differences were considered significant when the $p$-value was < 0.05. Comparisons among 3 or more groups were performed using 1-way ANOVA, followed by multiple comparison test. (*, $p < 0.05$; **, $p < 0.01$; ***, $p < 0.001$; NS, not significant). Statistical analyses were performed using Graph-Pad Prism v.5 (GraphPad Software, USA).

## Supporting information

**S1 Fig. YAP/TAZ expression is activated by pro-inflammatory or reparative macrophage-stimuli in wild type BMDMs.** BMDMs were isolated from wild-type mice and stimulated with/without LPS/IFNγ or IL4/IL13 for 0, 8 and 12 hours, respectively. Western blot analysis for YAP and TAZ was performed using the nuclear fraction of BMDMs. PCNA is shown as a loading control. The relative expression was quantified. Western blot analysis for pYAP was performed using the cytoplasmic fraction of BMDMs. B-actin is shown as a loading control. The relative expression was quantified. For numerical raw data, please see S1 Data.
(PDF)

**S2 Fig. Characterization of BMDMs polarization into pro-inflammatory and reparative macrophages.** (A) BMDMs and PMs were isolated from wild-type mice and stimulated with/without LPS/IFNγ for 0, 4, 8, and 12 hours, respectively. Western blot analysis for TGFβ using cell lysates of BMDMs and PMs. Vinculin is shown as a loading control. The relative expression was quantified. (B) BMDMs and PMs were isolated from wild-type mice and stimulated with/without LPS/IFNγ for 12 hours. Real-time qPCR for reparative marker genes, *Vegf* and *Tgfβ*, using RNA isolated from untreated or LPS/IFNγ-treated macrophages. (C) BMDMs were isolated from wild-type mice and stimulated with/without IL4/IL13 for 0, 4, 8, and 12 hours, respectively. Western blot analysis for IL6 using cell lysates of BMDMs. Vinculin is shown as a loading control. The relative expression was quantified. For numerical raw data, please see S1 Data.
(PDF)

**S3 Fig. Knockdown of *TAZ* decreases pro-inflammatory and increases reparative macrophage phenotype.** (A–C) BMDMs were isolated from wild-type mice and transfected with control or *TAZ* siRNA for 72 hours, followed by LPS/IFNγ or IL4/IL13 stimulation for 16

hours. Cell lysates were prepared for western blot and qRT-PCR analysis. (A) Western blot analysis for TAZ, IL6, and Nos2 was performed using total lysates from wild-type BMDMs transfected with control or TAZ siRNA. Vinculin is shown as a loading control. The relative expression was quantified. (B) Real-time qPCR for pro-inflammatory marker genes *IL6*, *IL1β*, and *Nos2* using RNA isolated from wild-type BMDMs transfected with control or *TAZ* siRNA and stimulated with LPS/IFNγ. (C) Real-time qPCR for reparative marker genes *Arg1*, *Ym1*, and *Fizz1* using RNA isolated from wild-type BMDMs transfected with control or *TAZ* siRNA and stimulated with IL4/IL13. (D) Real-time qPCR for *IL6*, *IL1β*, and *Nos2* using RNA isolated from BMDMs treated with either DMSO, LPS/IFNγ, or LPS/IFNγ together with verteporfin (VP). Data are shown as the mean ± SEM, *n* = 3 for each experimental group. Gene expression data were normalized with the reference gene *Gapdh*, and results are represented as fold change relative to the control treatment. For numerical raw data, please see S1 Data. (PDF)

**S4 Fig. Efficiency of YAP/TAZ deletion in macrophages.** (A) Real-time qPCR for *YAP* and *TAZ* using RNA isolated from control and *YAP/TAZ*-dKO BMDMs. *n* = 3 in each group. (B) Western blot analysis for YAP and TAZ was performed using total lysates from control and *YAP/TAZ-dKO* BMDMs. Vinculin is shown as a loading control. For numerical raw data, please see S1 Data. (PDF)

**S5 Fig. Characterization of isolated BMDMs and PMs.** (A and B) Isolated BMDMs (A) and PMs (B) were stained with CD11b-FITC and APC anti-mouse F4/80 antibodies, and their purity were evaluated by flow cytometry analysis. (C and D) Representative flow cytometry analysis of pro-inflammatory (iNOS$^+$/F4/80$^+$) and reparative (CD206$^+$/F4/80$^+$) macrophages in total macrophages from the 2 and 6 days post-MI hearts, respectively; showing the percentage of F4/80$^+$iNOS$^+$ and F4/80$^+$CD206$^+$ macrophages in *YAP/TAZ-dKO* hearts compared to respective control. (TIF)

**S6 Fig. BMDMs proliferation and migration is not affected due to *YAP/TAZ* deletion.** (A) To assess changes in BMDMs proliferation, cultured BMDMs were stimulated with LPS/IFNγ and immunostained with CD68 and Ki67. DAPI was used to stain nuclei. Scale bar 100 μM. (B) Quantification of CD68 and Ki67 double-positive BMDMs from control and *YAP/TAZ*-dKO mice. (C) Migration of control and *YAP/TAZ*-dKO BMDMs was examined by a transwell assay. Cells that had migrated to the lower chamber of the transwell plate were visualized by CD68 immunostaining. DAPI was used to stain nuclei. Scale bar 50 μM. (D) Quantification of CD68 positive BMDMs from control and *YAP/TAZ*-dKO mice migrated to the lower chamber of the transwell plate. For numerical raw data, please see S1 Data. (PDF)

**S7 Fig. Changes in inflammatory gene expression due to *YAP/TAZ* deletion.** MA plots for pathways identified from pathway enrichment analysis of the RNA-seq data from untreated or LPS/IFNγ treated control and *YAP/TAZ*-dKO BMDMs. For numerical raw data, please see S1 Data. (PDF)

**S8 Fig. Effect of YAP on the expression of inflammatory genes.** Two luciferase reporters (*NF-kB-Luc* and *IL6-Luc*) were used with YAP in the presence or absence of Hippo signaling inhibitor verteporfin (VP) or MAPK inhibitor SB203580. (A) Results of normalized luciferase reporter assays in HEK293T cells with *NF-κB*-luciferase reporter in the presence of YAP. (B)

*IL6*-luciferase reporters were transfected in HEK293T cells with or without YAP in the presence or absence of VP or SB203580. All the experiments repeated at least 3 times. For numerical raw data, please see S1 Data.
(PDF)

**S9 Fig. Expression of reparative macrophage subset marker genes in control and *YAP/TAZ-dKO* BMDMs.** (A–C) BMDMs were isolated from control and *YAP/TAZ*-dKO mice and stimulated with/without IL4/IL13 (M2a) or BSA/anti-BSA immune complex (M2b) or IL10/TGFβ1 (M2c) for 24 hours. Real-time qPCR was performed for (A) M2a (*Ym1* and *Cd206*), (B) M2b (*Il10* and *Il1ra*), and (C) M2c (*Mmp9* and *TGFβ*) marker genes using RNA isolated from control and *YAP/TAZ*-dKO BMDMs. Data are shown as the mean ± SEM, *n* = 3 for each experimental group. Gene expression data were normalized with the reference gene *Gapdh*, and results are represented as fold change relative to the control treatment. For numerical raw data, please see S1 Data.
(PDF)

**S10 Fig. p53 expression was not affected in *YAP/TAZ-dKO* BMDMs during pro-inflammatory or reparative macrophage polarization.** (A) Relative expression (FPKM) of *Trp53* from RNA-seq analysis on control and *YAP/TAZ*-dKO BMDMs treated with LPS/IFNγ. (B and C) Western blot analysis and quantification for p53 were performed using total lysates from control and *YAP/TAZ-dKO* BMDMs stimulated with IL4/IL13 for 24 hours. β-actin is shown as a loading control. For numerical raw data, please see S1 Data.
(PDF)

**S11 Fig. Tead-binding sites in *Arg1* promoter.** (A) Reported-Tead binding sequences (TBSs). (B) Predicted TBSs in *Arg1* promoter.
(PDF)

**S12 Fig. *YAP/TAZ*-deficient macrophages exhibit reduced bactericidal activity.** (A) Experimental design of the bacterial killing assay. Briefly, BMDMs were infected with gram-negative (*E.coli*) or gram-positive (*L. monocytogenes*) bacteria for 2 hours and after thoroughly washed, incubated for 24 hours to determine the bactericidal activity of BMDMs derived from control mice against *YAP/TAZ*-deficient mice. (B and C) *YAP/TAZ*-deficient BMDMs show reduced bactericidal activity against *E. coli* ex vivo. (D) Nitric oxide (NO) production determined by nitrite (NO$_2$−) levels in conditioned medium of uninfected or *E.coli* infected control and *YAP/TAZ*-deficient BMDMs. (E) The expression of pro-inflammatory genes involved in bactericidal activity was decreased in YAP/TAZ-deficient BMDMs after incubation with *E. coli*. *n* = 3 in each group. (F and G) *YAP/TAZ*-deficient BMDMs exhibit reduced bactericidal activity against *L. monocytogenes* ex vivo. (H) Nitrite (NO$_2$−) levels in conditioned medium of uninfected or *L. monocytogenes* infected control and *YAP/TAZ*-deficient BMDMs. (I) The expression of pro-inflammatory genes involved in bactericidal activity was decreased in *YAP/TAZ*-deficient BMDMs after incubation with *L. monocytogenes*. For numerical raw data, please see S1 Data.
(PDF)

**S13 Fig. Macrophage-specific YAP overexpression exhibit enhanced bactericidal activity.** (A) Experimental design of the bacterial killing assay. Briefly, BMDMs were infected with gram-negative (*E.coli*) or gram-positive (*L. monocytogenes*) bacteria for 2 hours and after thoroughly washed, incubated for 24 hours to determine the bactericidal activity of BMDMs derived from control and *YAP$^{5SA}$* mice. (B and C) YAP5SA BMDMs show increased bactericidal activity against *E. coli* and *L. monocytogenes* ex vivo, respectively. (D) NO production by

BMDMs was determined by nitrite ($NO_2-$) levels in conditioned medium. (E) The expression of pro-inflammatory genes involved in bactericidal activity was enhanced in YAP5SA BMDMs after incubation with *E. coli* or *L. monocytogenes*. $n = 3$ in each group. For numerical raw data, please see S1 Data.
(PDF)

**S14 Fig. Cardiac functional defects in control and *YAP/TAZ*-deficient mice 4 weeks post-MI.** Echocardiographic measurements of ventricular functional parameters at baseline and 4 weeks post-MI from control and *YAP/TAZ*-dKO mice: body weight (BW); and left ventricular mass (LV Mass); left ventricular end-diastolic volume (LVEDV); and stroke volume (SV). Data are shown as mean ± SEM ($n = 4$ to 7 per group). For numerical raw data, please see S1 Data.
(PDF)

**S15 Fig. Neutrophil recruitment and migration are not affected due to *YAP/TAZ* deletion.** (A) Immunohistochemistry and quantification for MPO on control and *YAP/TAZ-dKO* infarcted heart sections at 2 days post-MI. The number of MPO positive cells was determined in ≥5 distinct microscope fields for each heart section. At least 4 hearts were analyzed for each group. Scale bar represents 100 μm. (B) Migration of control and *YAP/TAZ*-dKO neutrophils was examined by a transwell assay. Cells that had migrated to the lower chamber of the transwell plate were visualized by MPO immunostaining. DAPI was used to stain nuclei. Scale bar 100 μM. Quantification of MPO positive neutrophils from control and *YAP/TAZ*-dKO mice migrated to the lower chamber of the transwell plate. For numerical raw data, please see S1 Data.
(PDF)

**S16 Fig. Hypoxia induces YAP/TAZ expression in BMDMs.** (A) BMDMs were isolated from wild-type mice and exposed to hypoxia with $H_2O_2$ (100 μM) for 8 hours. Western blot analysis for YAP, TAZ, and Nos2 was performed using total lysates from unstimulated and $H_2O_2$-stimulated cells. Vinculin is shown as a loading control. (B) The relative protein expression of YAP, TAZ, and Nos2 was quantified. For numerical raw data, please see S1 Data.
(PDF)

**S17 Fig. Control and *YAP/TAZ-dKO* mice demonstrate equal size ischemic injury region at 2 days post-MI.** (A) Whole-mount images of control and mutant animals as indicated, 2 days post-MI. The white dotted line indicates the area of injury near the ligature. Scale bar 500 μM. Quantification of Ischemic area, ($n = 3$ per group). (B) Masson's trichrome staining and quantification of fibrosis on control and *YAP/TAZ-dKO* heart at 2 days post-MI. The area was measured in 5≥ distinct microscope field, and the fibrotic area was normalized to the remaining heart for each heart section. At least 3 hearts were analyzed for each group. For numerical raw data, please see S1 Data.
(PDF)

**S18 Fig. *YAP/TAZ* deletion leads to reduced interstitial fibrosis and hypertrophic response post-MI.** (A and B) Masson's trichrome staining and quantification of interstitial myocardial fibrosis on control and *YAP/TAZ-dKO* heart sections at 28 days post-MI. The area was measured in 5≥ distinct microscope field, and the fibrotic area was normalized to the remaining heart for each heart section. At least 4 hearts were analyzed for each group. Scale bar represents 100 μm. (C and D) The cross-sectional area of the infarcted heart (border and remote zone) from control and *YAP/TAZ* dKO mice at 28 days post-MI. The area was measured in ≥5 distinct microscope fields for each heart section. At least 3 hearts were analyzed for each group.

Scale bar represents 50 μm. (E) The expression of genes such as *Myh7*, *Nppa*, and *Nppb* involved in hypertrophy was reduced in *YAP/TAZ*-dKO heart compared to control at 28 days post-MI. *n* = 3 in each group. For numerical raw data, please see S1 Data.
(PDF)

**S19 Fig. Improved revascularization in *YAP/TAZ-dKO* hearts post-MI.** (A) Immunohistochemistry and quantification for CD31 on control and *YAP/TAZ-dKO* heart sections at 28 days post-MI. The area was measured in ≥3 distinct microscope fields for each heart section. At least 3 hearts were analyzed for each group. Scale bar represents 100 μm. (B) Increased capillary density in *YAP/TAZ-dKO* heart at 28 days post-MI along with the corresponding quantification. At least 3 hearts were analyzed for each group. Scale bar 50 μM. For numerical raw data, please see S1 Data.
(PDF)

**S20 Fig.** FACS data for proinflammatory macrophage isolation from heart 2 days port-MI. Gating strategies used for isolation of pro-inflammatory (iNOS+/F4/80+) macrophages from the 2 days post-MI hearts.
(PDF)

**S21 Fig.** FACS data for reparative macrophage isolation from heart 6 days port-MI. Gating strategies used for isolation of reparative (CD206+/F4/80+) macrophages from the 6 days post-MI hearts.
(PDF)

**S1 Data. Excel spreadsheet with numerical raw data underlying Fig 1A, 1B, 1C, 1D, 1E; Fig 2C, 2D, 2E; Fig 3A, 3B, 3E, 3G, 3F, 3H; Fig 4A, 4B, 4C, 4D, 4E, 4F, 4G, 4H, 4I, 4J, 4K, 4N, 4O, 4P, 4Q, 4R; Fig 5A, 5B, 5C, 5D, 5F, 5G, 5H, 5I, 5J; Fig 6B, 6D, 6E, 6F, 6G; Fig 7D, 7E, 7F, 7H, 7J; S1 Fig; S2A, S2B, S2C Fig; S3A, S3B, S3C, S3D Fig; S4A Fig; S6B, S6D Fig; S7 Fig; S8A, S8B, S8C Fig; S9A, S9B, S9C Fig; S10A, S10B Fig; S12C, S12D, S12E, S12G, S12H, S12I Fig; S13B, S13C, S13D, S13E Fig; S14 Fig; S15A, S15B Fig; S16B Fig; S17A, S17B Fig; S18B, S18D, S18E Fig and S19A, S19B Fig.**
(XLSX)

**S1 FCS file. FCS file for Fig 5F_ pro-inflammatory macrophages_control 1.**
(FCS)

**S2 FCS file. FCS file for Fig 5F_ pro-inflammatory macrophages_control 2.**
(FCS)

**S3 FCS file. FCS file for Fig 5F_ pro-inflammatory macrophages_control 3.**
(FCS)

**S4 FCS file. FCS file for Fig 5F_ pro-inflammatory macrophages_dKO 1.**
(FCS)

**S5 FCS file. FCS file for Fig 5F_ pro-inflammatory macrophages_dKO 2.**
(FCS)

**S6 FCS file. FCS file for Fig 5F_ pro-inflammatory macrophages_dKO 3.**
(FCS)

**S7 FCS file. FCS file for Fig 5F_ pro-inflammatory macrophages_dKO 4.**
(FCS)

**S8 FCS file. FCS file for Fig 5F_ pro-inflammatory macrophages_DAPI.**
(FCS)

**S9 FCS file. FCS file for Fig 5F_ pro-inflammatory macrophages_unstained.**
(FCS)

**S10 FCS file. FCS file for Fig 5F_ reparative macrophages_control 1.**
(FCS)

**S11 FCS file. FCS file for Fig 5F_ reparative macrophages_control 2.**
(FCS)

**S12 FCS file. FCS file for Fig 5F_ reparative macrophages_control 3.**
(FCS)

**S13 FCS file. FCS file for Fig 5F_ reparative macrophages_control 4.**
(FCS)

**S14 FCS file. FCS file for Fig 5F_ reparative macrophages_control 5.**
(FCS)

**S15 FCS file. FCS file for Fig 5F_ reparative macrophages_dKO 1.**
(FCS)

**S16 FCS file. FCS file for Fig 5F_ reparative macrophages_dKO 2.**
(FCS)

**S17 FCS file. FCS file for Fig 5F_ reparative macrophages_dKO 3.**
(FCS)

**S18 FCS file. FCS file for Fig 5F_ reparative macrophages_dKO 4.**
(FCS)

**S19 FCS file. FCS file for Fig 5F_ reparative macrophages_dKO 5.**
(FCS)

**S20 FCS file. FCS file for Fig 5F_ reparative macrophages_DAPI.**
(FCS)

**S21 FCS file. FCS file for Fig 5F_ reparative macrophages_unstained.**
(FCS)

**S22 FCS file. FCS file for Fig 5F_ reparative macrophages_isotype.**
(FCS)

## Acknowledgments

We are thankful to M.K.S. lab members for helpful discussion. We thank Sindhu Ramesh for technical support. We thank Dr. Chinmay Trivedi (University of Massachusetts Medical School) for providing HDAC3 and HDAC3$^{H134A, H135A}$ plasmids.

## Author Contributions

**Conceptualization:** Manvendra K. Singh.

**Data curation:** Masum M. Mia, Dasan Mary Cibi, Siti Aishah Binte Abdul Ghani, Weihua Song, Nicole Tee, Sujoy Ghosh.

**Formal analysis:** Masum M. Mia, Dasan Mary Cibi, Siti Aishah Binte Abdul Ghani, Nicole Tee, Sujoy Ghosh, Manvendra K. Singh.

**Funding acquisition:** Manvendra K. Singh.

**Investigation:** Masum M. Mia, Dasan Mary Cibi, Manvendra K. Singh.

**Methodology:** Masum M. Mia, Dasan Mary Cibi, Siti Aishah Binte Abdul Ghani, Weihua Song, Nicole Tee, Sujoy Ghosh, Manvendra K. Singh.

**Project administration:** Manvendra K. Singh.

**Resources:** Junhao Mao, Eric N. Olson, Manvendra K. Singh.

**Software:** Sujoy Ghosh.

**Supervision:** Masum M. Mia, Manvendra K. Singh.

**Validation:** Masum M. Mia, Dasan Mary Cibi, Siti Aishah Binte Abdul Ghani, Weihua Song, Nicole Tee.

**Visualization:** Siti Aishah Binte Abdul Ghani.

**Writing – original draft:** Masum M. Mia.

**Writing – review & editing:** Manvendra K. Singh.

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
