## [Editor Report · Decision Letter 0]

31 Oct 2019

Dear Dr Singh, 

Thank you for submitting your manuscript entitled "YAP/TAZ deficiency reprograms macrophage phenotype and improves infarct healing and cardiac function after myocardial infarction" for consideration as a Research Article by PLOS Biology.

Your manuscript has now been evaluated by the PLOS Biology editorial staff as well as by an academic editor with relevant expertise and I am writing to let you know that we would like to send your submission out for external peer review. 

Please re-submit your manuscript within two working days, i.e. by Nov 02 2019 11:59PM.

Kind regards,

Di Jiang, PhD

Associate Editor

PLOS Biology

---

## [Decision Letter · Decision Letter 1]

27 Nov 2019

Dear Dr Singh,

Thank you very much for submitting your manuscript "YAP/TAZ deficiency reprograms macrophage phenotype and improves infarct healing and cardiac function after myocardial infarction" for consideration as a Research Article at PLOS Biology. Your manuscript has been evaluated by the PLOS Biology editors, an Academic Editor with relevant expertise, and by independent reviewers.

The reviews of your manuscript are appended below. You will see that the reviewers find the work potentially interesting. However, based on their specific comments and following discussion with the academic editor, I regret that we cannot accept the current version of the manuscript for publication. We remain interested in your study and we would be willing to consider resubmission of a comprehensively revised version that thoroughly addresses all the reviewers' comments. We cannot make any decision about publication until we have seen the revised manuscript and your response to the reviewers' comments. Your revised manuscript would be sent for further evaluation by the reviewers.

Of particular note, the reviewers note small effect sizes and question whether these are sufficient to lead to a biological effect. Additionally, Rev #1 and the Academic Editor believe that the M1 versus M2 dichotomy is an outdated concept and require a more nuanced examination of phenotypes.

We appreciate that these requests represent a great deal of extra work, and we are willing to relax our standard revision time to allow you six months to revise your manuscript. Please email us (plosbiology@plos.org) to discuss this if you have any questions or concerns, or think that you would need longer than this. At this stage, your manuscript remains formally under active consideration at our journal; please notify us by email if you do not wish to submit a revision and instead wish to pursue publication elsewhere, so that we may end consideration of the manuscript at PLOS Biology.

Your revisions should address the specific points made by each reviewer. Please submit a file detailing your responses to the editorial requests and a point-by-point response to all of the reviewers' comments that indicates the changes you have made to the manuscript. In addition to a clean copy of the manuscript, please upload a 'track-changes' version of your manuscript that specifies the edits made. This should be uploaded as a "Related" file type. You should also cite any additional relevant literature that has been published since the original submission and mention any additional citations in your response. 

Before you revise your manuscript, please review the following PLOS policy and formatting requirements checklist PDF: http://journals.plos.org/plosbiology/s/file?id=9411/plos-biology-formatting-checklist.pdf. It is helpful if you format your revision according to our requirements - should your paper subsequently be accepted, this will save time at the acceptance stage.

Please note that as a condition of publication PLOS' data policy (http://journals.plos.org/plosbiology/s/data-availability) requires that you make available all data used to draw the conclusions arrived at in your manuscript. If you have not already done so, you must include any data used in your manuscript either in appropriate repositories, within the body of the manuscript, or as supporting information (N.B. this includes any numerical values that were used to generate graphs, histograms etc.). For an example see here: http://www.plosbiology.org/article/info%3Adoi%2F10.1371%2Fjournal.pbio.1001908#s5.

For manuscripts submitted on or after 1st July 2019, we require the original, uncropped and minimally adjusted images supporting all blot and gel results reported in an article's figures or Supporting Information files. We will require these files before a manuscript can be accepted so please prepare them now, if you have not already uploaded them. Please carefully read our guidelines for how to prepare and upload this data: https://journals.plos.org/plosbiology/s/figures#loc-blot-and-gel-reporting-requirements.

Upon resubmission, the editors will assess your revision and if the editors and Academic Editor feel that the revised manuscript remains appropriate for the journal, we will send the manuscript for re-review. We aim to consult the same Academic Editor and reviewers for revised manuscripts but may consult others if needed.

If you still intend to submit a revised version of your manuscript, please go to https://www.editorialmanager.com/pbiology/ and log in as an Author. Click the link labelled 'Submissions Needing Revision' where you will find your submission record. 

Sincerely,

Lauren Richardson, Ph.D

Senior Editor

on behalf of 

Di Jiang, Ph.D

Associate Editor

PLOS Biology

Reviews

Reviewer #1: 

In the submitted manuscript, Mia et al. investigate the functions of YAP/TAZ in myeloid cells following myocardial infarction. Using a bone marrow-derived macrophage model of macrophage differentiation, the authors provide evidence that YAP/TAZ are necessary and sufficient for optimal expression of genes associated with inflammatory responses (M1) and suppression of genes associated with a reparative (M2) macrophage phenotype. Mechanistically YAP/TAZ promoted inflammatory gene expression by directly binding to the IL-6 promoter and augmenting TLR4 signaling through a MAPK dependent pathway and impaired Arg1 expression (M2 marker) by interacting with the HDAC3-NCoR1 repressor complex. The authors explored the roles of YAP/TAZ in myeloid cells using an experimental mouse model of coronary ligation. Loss of YAP/TAZ in myeloid cells resulted in improved LV systolic function, reduced fibrosis, and increased coronary angiogenesis. Conversely, YAP activation led to exaggerated post-MI LV remodeling and increased myocardial fibrosis. The authors provide further evidence that manipulation of hippo signaling in myeloid cells influenced macrophage gene expression.

Overall, this manuscript is well written and the data are clearly presented. The finding that YAP/TAZ influence macrophage gene expression and potentially monocyte differentiation are novel and interesting. However, several conceptual and methodologic issues limit the validity of the author’s conclusions and impact of this study.

Specific comments

1. What is the biologically relevant signal that controls activation of YAP/TAZ in vivo? The authors provide data that macrophage activating stimuli (LPS/IFNg and IL4/13) result in increased YAP and TAZ mRNA and protein expression. Do the authors believe that modulation of YAP/TAZ expression is sufficient to control hippo signaling? As YAP/TAZ exert their effects within the nucleus, the authors should consider measuring nuclear YAP/TAZ protein abundance using fractionated cell preparations (nucleus vs. cytoplasm). The authors also comment on increased in phosphorylated YAP. It would be more ideal to measure non-phosphorylated YAP, which is the active form within the nucleus.

2. The authors conclude that YAP/TAZ potentiate inflammatory macrophage gene expression through two potential mechanisms: directly binding to the IL-6 promoter and augmentation of TLR4/TAK1/MAPK signaling. What is the relative contribution of each of these mechanisms? Does YAP/TAZ bind to the promoter of other inflammatory chemokines and cytokines that are regulated by YAP/TAZ? Does activated YAP potentiate TLR4 signaling in a TAK1/MAPK dependent manner? Functional data implicated the relative importance of these pathways are lacking. What is the mechanism by which YAP/TAZ influences TLR4/TAK1/MAPK signaling?

3. The authors conclude that YAP/TAZ inhibits Arginase1 expression through an AKT dependent pathway that involves recruitment of the HDAC3-NCoR1 repressor complex to the arginase promoter. Functional data implicating AKT signaling is lacking. Does inhibition of AKT prevent assembly of this repressor complex? What is the mechanism by which YAP/TAZ deficiency augments AKT signaling? The HDAC3 siRNA experiments are not informative. A more ideal experiment would be to prevent the interaction between YAP and HDAC3.

4. The experiment myocardial infarction studies show clear and consistent phenotypes. However, the use of Lyz2-Cre is an important limitation as it is active in all myeloid cells including granulocytes. Thus, it is not possible to conclude that YAP/TAZ deficiency in monocytes and macrophages is responsible for the observed phenotypes. As many neutrophils are recruited to the heart following myocardial infarction, the authors should either use more selective Cre recombinases or provide evidence as to whether YAP/TAZ influence neutrophil phenotypes.

5. Presented echocardiographic data is limited to ejection fraction and fractional shortening. Given the asymmetric pattern of injury and left ventricular (LV) remodeling, fractional shortening is not an ideal measurement. The authors should include quantitative data pertaining to LV ejection fraction, LV diastolic and systolic volumes, and LV mass. 

6. To assess whether changes in infarct size are a result of increased initial infarct area or infarct expansion, the authors should perform TTC staining within 24-48 hours of myocardial infarction to assess initial infarct size in each experimental group.

7. Pathological analysis of LV remodeling is limited to measurements of infarct size and coronary capillary density. The authors should also include quantification of cardiomyocyte cross sectional area within the border and remote zones as well as interstitial myocardial fibrosis.

8. A key conceptual limitation of this manuscript is the focus on M1 vs. M2 macrophages. This is an outdated classification of macrophages that is derived from in vitro polarization studies. A more modern view is that monocytes have the capacity to differentiate into a variety of macrophage subtypes with phenotypes that range between the extremes formulated in the M1 and M2 designations. Markers of in vitro derived M1 and M2 macrophages rarely coincide with in vivo populations. Furthermore, the heart contains both resident macrophages and recruited monocyte-derived macrophages that are not described by the M1/M2 classification. While the authors mRNA expression data evaluating macrophage gene expression are informative, the quantification of NOS2+ and CD206+ macrophages is less helpful. Differences between CD206+ macrophages seem to be driven by 2 outlying data points. If the authors wish to make rigorous conclusion regarding monocyte and macrophage differentiation, it would be more helpful to partition macrophages into established subsets such as CCR2-/LYVE1+ and CCR2+/LYVE1- populations or use single cell RNA sequencing.

----------------

Reviewer #2: 

In their paper, Mia et al. show how differentiation of bone marrow-derived macrophages (BMDMs) to pro-inflammatory (M1) or anti-inflammatory (M2) macrophages is regulated in vitro and after myocardial infarction (MI). After MI, ineffective recovery and adverse cardiac remodelling initiate structural and also functional changes in the heart and are the cause of developing heart failure. Thereby, a complex inflammation cascade regulates the initial pro-inflammatory M1-mediated response as well as the following anti-inflammatory response, which is mediated by M2 macrophages and supposed to be the reparative phase. The rational of this study was to find new regulators of macrophage polarization and post-MI inflammation.

The authors concentrated on YAP and TAZ, two central transcriptional co-factors of the Hippo signalling pathway. They can show that YAP/TAZ expression is increased in BMDMs during M1 or M2 polarization and that in YAP/TAZ-dKO mice (Myeloid specific YAP/TAZ knock-out), M1 polarization is impaired while M2 polarization is favoured leading to reduced fibrosis, increased angiogenesis and improved heart function 28d post-MI. This is mediated by interaction of YAP with the IL-6 promotor and the HDAC3-NCoR1 repressor complex. These results may render YAP/TAZ as future therapeutically targets in enhancing the anti-inflammatory response after MI. The study is well conducted and interesting, although the shown interaction of YAP/TAZ with the Hippo signalling pathway is not novel (as already mentioned in the references; Zhou et al., 2019). The novelty is the finding that the same mechanism is responsible for macrophage differentiation after MI and the YAP/TAZ interaction with HDAC3-NCoR1 complex.

In addition, some questions remain.

Specific points:

• Fig. 1A and B: The proposed activation of YAP after LPS/IFNγ stimulation by enhanced phosphorylation of S127 is hard to see in the depicted western blots and blots either should be changed or removed (Fig. 1A). In addition, the increase in IL-6 expression in BMDMs is not convincing. The exposure of the blot showing Actin expression in PMs is too long and should be reduced. Expression of IL-10, VEGF or TGF-ß should be shown to exclude differentiation to M2 macrophages. Please also include quantification of all Western Blots. 

• Fig. 1C: Results should be shown at least in duplicates and be quantified. There is still a significant expression of IL-6 in YAP depleted BMDMs. Does that mean a subpopulation of BMDMs nevertheless differentiated to M1 macrophages? This is also confirmed by the reduction of M1 marker gene expression by only around 20% compared to control siRNA with stimulation (Fig. 1D). Western blot data for Control siRNA without stimulation should be included. In general, do the authors have data regarding siRNA mediated TAZ downregulation in BMDMs or data showing effects of YAP/TAZ downregulation in PMs? Or are the observed effects restricted to BMDMs and circulating macrophages, respectively?

• Fig. 1E: It would be interesting to see protein expression data of IL-6 and IL-10, VEGF or TGF-ß. This could further help to evaluate the differentiation pattern.

• Supp. Fig. 4 and 5: Results showing data of macrophages in an activated state would strengthen the hypothesis that Yap/TAZ-dKO did not affect proliferation or migration properties of BMDMs. 

• Fig. 3 and 4: Of much interest would be the effect of YAP/TAZ-dKO in BMDMs on proliferation and migration. This could include a rescue experiment e.g. with recombinant IL-6.

• Fig. 3B and C: The reduced protein expression of P-Tak1 and of the key components of MAPKs is only seen after 15min of LPS/INFγ treatment. 30min after stimulation expression pattern are equal compared to control. Furthermore, since phosphorylation is a posttranslational modification, it cannot be measured with qRT-PCR. Here, total mRNA expression is analysed and again only 15min after stimulation there is a difference in mRNA expression. That could explain the increase in phosphorylated protein in that time point. That stresses the point that YAP/TAZ-dKO in BMDMs affects a mechanisms or signalling pathway upstream of MAPKs and Tak1 axis. Since total mRNA expression is also affected as seen in the qRT-PCR results western blot data should be included in Fig. 3B.

• Fig. 3: Do the authors have data regarding a direct promotor binding of other M1 marker genes depicted in Fig. 1D? Does YAP only binds IL-6 promotor directly? How about TAZ?

• Fig. 4B and C: Actin blots are exposed too long. Western blots should be quantified.

• Fig. 4I: Please also include data for WB: Myc and the respective band sizes in all blots. Are data available showing IP: Flag with WB: Myc?

• Fig. 5 and 6: It is not clear whether the infarct region or remote region is the focus of investigation here. Please indicate.

• Fig. 5E: There is a huge variance in the macrophage subtype population 6d post MI in the YAP/TAZ-dKO group. Do the authors have any explanation for that? Please also include flow cytometry analysis of M1 and M2 macrophages of all in Fig. 5F-I indicated time points.

• Fig. 6A and 7G: How was the measured fibrotic area normalized? Normalized to the remaining heart or to the left ventricle? Furthermore, all heart section should be analysed at the level of the papillary muscle. This is not evident by the depicted pictures.

• Fig. 7G: Please also include echo data.

• Please also conduct and include an early echocardiography time point in Figure 5 and Figure 7 to exclude (or show) an initial difference in infarct size between Yap/Taz knock-out, control and YAP overexpressing mice. Alternatively, Serum Troponin levels could be determined to demonstrate similar initial myocardial injury.

----------------

Reviewer #3: 

Using a range of elegant techniques that span, the authors show differential regulation of M1 and M2 macrophages by YAP/TAZ. Specifically, YAP/TAZ is required for M1 polarisation whereas it is inhibitory to M2 activation.

The authors identify the targets of YAP/TAZ in these different macrophage states and identify that Ncor/HDAC repressor complex participates in the inhibition of gene expression by YAP. 

Notably, through decreasing M1 macrophages, YAP/TAZ deletion enhances post-MI remodelling and reduces fibrosis. 

This story is generally sound and the data is internally consistent. The findings provided add to the increasing knowledge of the YAP/TAZ pathway in the heart as well as other organs. These findings are of significance and contribute to our knowledge of how M1 and M2 macrophages are regulated in the heart. 

I have a few issues that require addressing. 

1. The authors rule out the NFkB pathway as a mechanism for activation of IL6 expression based on lack of phosphorylation of IKKɑβ. The sensitivity of this assay may not be sufficient to rule out its role. I would suggest analysis of a luciferase reporter of NfkB signalling. 

2. The effects of YAP/TAZ DKO on the expression of M1 markers revealed by qRT-PCR are not particularly dramatic. While these effects of YAP-TAZ KO are significant, in terms of absolute effects on induction they are quite small, cf. Rantes from a 300 to 200 fold induction? Yet, an end biological effect is reported – albeit small. Can the abundance of these cytokines be directly measured. How does this change in mRNA relate to protein? I have similar concerns re Fig 4A and Fig 7A. RE Fig 4A, the absolute changes in expression of M2 marker induction by YAP/TAZ KO are relatively small. Is the effect substantial enough to explain the biology proposed? In Fig 7 A, we see the effects of overexpression of a constitutively active YAP. Yet again, the relative changes in message are small.

3. These qRT-PCR data were normalised against one reference gene, GAPDH. Normalising to one reference gene is not best practice. Was this reference gene deemed to eb stable between all conditions analysed. If so, provide the data to show its expression relative to a panel of other reference genes. 

4. The authors indicate that that YAP/TAZ interacts with the HDAC3-NCoR1 complex to repress Arg1expression in macrophages. In support of this, they show that by IP experiments the interaction of the proteins in this complex and that YAP directly binds to the promoter of the ARg1. It was not clear to me whether experiments were conducted to test a direct interaction of YAP with DNA. The authors go on to show that the transfection of HDAC represses the expression of this reporter. The observation that this Luc reporter is transiently transfected and not chromatinised would make it surprising that the repressor complex was operating in such a manner. I do not rule out this possibility however, especially if the repression is not via alterations in histone acetylation but via a direct protein mediated repression. Experiments to identify the interaction domain between YAP and the repressor complex which could then be modified to prevent such an interaction would substantially increase the weight of evidence in favour of a functional interaction between YAP and the repressor complex. 

5. Subsequent experiments in which HDAC3 has been reduced in expression by siRNA may also provide misleading results. HDAC would of course repress the expression of many genes, some of which may not be related to the endpoints studied in this MS. Some of these effects will be via protein/protein interactions which will be disrupted by KD. I do not understand why supporting experiments using an inhibitor of HDAC were also not performed. These would rule in/out the role of HDAC activity and could contribute to the understanding of the structural vs enzymic role of HDAC in repression of Arg.

6. The authors provide data on EF and FS for heart function. When performing 2D echo, EF and FS are derived from the same measures and are thus not independent. The one reflects the other. Using the one measure to support the other is not appropriate. Based on both measures the effect of YAP KO are not substantial. It would be useful to have other measures of cardiac remodelling. For example, are there any differences in myocytes size, wall thickness, expression of fetal genes? Further and to support the contention that fibrosis is affected by YAP KO, a measure of strain or relaxation velocity would be useful. This would tell us the stiffness of the ventricle. 

7. IN Fig 6G the authors analyse the expression of endmucin in heart sections as a measure of capillary density. The data provided is a surface area coverage of this marker. It is not capillary density, which is the more often used measure. The authors should analyse for e,g. CD31 staining in tissue sections.

---

## [Decision Letter · Decision Letter 2]

17 Sep 2020

Dear Dr Singh,

Thank you for submitting your revised Research Article entitled "YAP/TAZ deficiency reprograms macrophage phenotype and improves infarct healing and cardiac function after myocardial infarction" for publication in PLOS Biology. Thank you again for your patience as we completed our editorial process. I have now obtained advice from the three original reviewers and have discussed their comments with the Academic Editor. 

Based on the reviews, we will probably accept this manuscript for publication, assuming that you will modify the manuscript to address the remaining points raised by Reviewers 1 and 3 - a few minor text changes (see comments pasted below). Please also make sure to address the data and other policy-related requests noted at the end of this email.

We expect to receive your revised manuscript within two weeks. Your revisions should address the specific points made by each reviewer. In addition to the remaining revisions and before we will be able to formally accept your manuscript and consider it "in press", we also need to ensure that your article conforms to our guidelines. A member of our team will be in touch shortly with a set of requests. As we can't proceed until these requirements are met, your swift response will help prevent delays to publication.

- a cover letter that should detail your responses to any editorial requests, if applicable

*Copyediting*

*Published Peer Review History*

*Early Version*

Best wishes,

Ines

--

Ines Alvarez-Garcia, PhD,

Senior Editor,

ialvarez-garcia@plos.org,

PLOS Biology

ETHICS STATEMENT:

-- Thank you for including your ethics statement. Please include the project license and/or approval number.

DATA POLICY:

Fig. 1A, B, C, D, E; Fig. 2C, D, E; Fig. 3A, B, E, G, F, H; Fig. 4A, B, C, D, E, F, G, H, I, J, K, N, O, P, Q, R; Fig. 5A, B, C, D, F, G, H, I, J; Fig. 6B, D, E, F, G; Fig. 7D, E, F, H, J; Fig. S1; Fig. S2A, B, C; Fig. S3A, B, C, D; Fig. S4A; Fig. S6B, D; Fig. S7; Fig8A, B, C; Fig. 9A, B, C; Fig. 10A, B; Fig. 12C, D, E, G, H, I; Fig. S13B, C, D, E; Fig. S14; Fig. S15A, B; Fig. S16B; Fig. S17A, B; Fig. S18B, D, E and Fig. S19A, B

For figures containing FACS data (Fig. S5), we ask that you provide FCS files and a picture showing the successive plots and gates that were applied to the FCS files to generate the figure.

Please also provide the accession number for the raw RNAseq data set submitted to GEO.

Reviewers’ comments

Rev. 1:

I have no further questions. The authors have adequately addressed my concerns. I do believe that it is more appropriate to use the terms "pro-inflammatory" and "reparative" phenotypes as opposed to M1/M2.

Rev. 2:

I have no further comments. The authors adequately addressed my concerns.

Rev. 3:

The Authors have made a significant effort to address the comments raised by all of the reviewers. I am satisfied by the responses of the authors and by the additional experiments included. I particularly appreciate the experiments to investigate the enzymatic vs structural role of HDAC3.

I have a few very minor comments.

1. in Figure 6 D, the y axis is labelled Collagen 1 Production. The legend indicates that this is collagen 1 +ve area of the section. The label collagen 1 production is misleading. Please replace with Collagen 1 +ve area.

2. Could the authors indicate whether their Echo is 3D. If only 2D echo, all volumes are based based on 2 D dimensions that as one of the other referees indicated may not accurately reflect alterations due to MI - which may be asymmetric in nature.

Text:

Page 28 (page 4 of MS), Para 2, line 21. in the sentence: development but also for limiting the inflammatory and fibrotic response during post-MI recovery phage. The last word should be phase.

Page 38 (page 10 in MS), Para 1, line 9. in the sentence : Alternatively, a decreased in the expression of.... decreased should be replaced with decrease.

---

## [Editor Report · Decision Letter 3]

29 Oct 2020

Dear Dr Singh,

On behalf of my colleagues and the Academic Editor, Thomas C. Freeman, I am pleased to inform you that we will be delighted to publish your Research Article in PLOS Biology. 

PRODUCTION PROCESS

Before publication you will see the copyedited word document (within 5 business days) and a PDF proof shortly after that. The copyeditor will be in touch shortly before sending you the copyedited Word document. We will make some revisions at copyediting stage to conform to our general style, and for clarification. When you receive this version you should check and revise it very carefully, including figures, tables, references, and supporting information, because corrections at the next stage (proofs) will be strictly limited to (1) errors in author names or affiliations, (2) errors of scientific fact that would cause misunderstandings to readers, and (3) printer's (introduced) errors. Please return the copyedited file within 2 business days in order to ensure timely delivery of the PDF proof. 

If you are likely to be away when either this document or the proof is sent, please ensure we have contact information of a second person, as we will need you to respond quickly at each point. Given the disruptions resulting from the ongoing COVID-19 pandemic, there may be delays in the production process. We apologise in advance for any inconvenience caused and will do our best to minimize impact as far as possible.

EARLY VERSION

PRESS 

Kind regards,

Alice Musson

Publishing Editor, 

PLOS Biology

on behalf of

Ines Alvarez-Garcia,

Senior Editor

PLOS Biology